# GUI-AIMA: ALIGNING INTRINSIC MULTI-MODAL ATTENTION WITH A CONTEXT ANCHOR FOR GUI GROUNDING

## ABSTRACT

Graphical user interface (GUI) grounding is a key function of computer-use agents, which maps natural-language instructions to actionable screen regions. Existing approaches based on Multimodal Large Language Models (MLLMs) typically formulate it as a text-based coordinate generation task, yet directly generating precise coordinates from visual inputs remains challenging and computationally intensive. An intuitive way to implement GUI grounding is to *first select visual patches relevant to the instructions and then determine the precise click location within those patches*. Based on the observations that general MLLMs have native grounding capability, which is highly correlated with query-to-visual attentions, we propose GUI-AIMA, an attention-only and coordinate-free supervised fine-tuning framework for efficient GUI grounding. This framework aligns the intrinsic multimodal attention of MLLMs with patch-wise grounding signals. Specifically, we convert coordinate-based grounding boxes into soft patch-wise labels considering patch overlap and the center click manner. For attention aggregation, we simplify the merging of attention predictions among all query tokens into a single anchored attention vector with learnable `<ANCHOR>` token. More importantly, GUI-AIMA includes a query-adaptive multi-head weighting mechanism for multi-head attention aggregation by prioritizing text-vision affinity heads with visual-sink query tokens. GUI-AIMA-3B, trained on a small training set only with 85k screenshots, achieves the state-of-the-art performance among 3B models (*i.e.*, **49.8%** average on ScreenSpot-Pro, **58.3%** average on OSWorld-G and **91.5%** average on ScreenSpot-v2).

## 1 INTRODUCTION

Graphical User Interface (GUI) (Hong et al., 2024; Cheng et al., 2024) agents have emerged as pivotal tools in automating interactions with digital devices, spanning mobile applications (Rawles et al., 2023; 2024; Wang et al., 2024a; Ye et al., 2025a) to desktop software (Zhang et al., 2024; Qin et al., 2025; Deng et al., 2023; Xie et al., 2024; OpenAI, 2025; Zheng et al., 2024). GUI grounding plays an important role in mapping natural language instructions to specific elements on the GUI screen, such as buttons, text fields, or icons (Yang et al., 2024). This process ensures that the agent's actions are precise and contextually relevant, especially in environments with high-resolution displays and intricate layouts. However, the diversity of human instruction and GUI designs across platforms and applications (Li et al., 2025a) makes GUI grounding a challenging task. Conventional methods rely on structured representations, such as HTML for web pages or accessibility trees (Zhang et al., 2025) for mobile applications. While these provide detailed information about the interface elements, they come with limited accessibility and verbosity, which can lead to inefficiencies in processing. Moreover, structured data may omit essential visual cues like layout and icons, which are critical for accurate grounding.

When humans use computers, they first determine the general area and then use visual feedback to interact with GUI elements (Ye et al., 2025b). Directly outputting coordinates from MLLMs might not be the most intuitive method. Instead, they should mimic human behavior: first selecting the relevant visual patch and then deciding the specific click position within that patch, *i.e.*, the coordinates in the GUI. Previous works, such as TAG (Xu et al., 2024a) and GUI-Actor (Wu et al.,

2025), have explored the coordinate-free manner of GUI grounding that treats it as selecting the most relevant visual patches in the screenshot conditioned on a natural-language instruction, instead of generating text-based bounding boxes. However, GUI-Actor introduces extra embedding-based grounding modules into the MLLM and requires extra adaptation stage, which is inefficient. And the vanilla multi-token aggregation in TAG for attention-style predictions is cumbersome, and the rough multi-head selection is biased for measuring the functionality of each head, restricting the real capacity of the attention-based visual grounding.

In this work, we propose GUI-AIMA, a coordinate-free and attention-only model for GUI grounding. The MLLM's multi-head self-attention (MHSA) (Vaswani et al., 2017) is trained to learn the patch-wise grounding supervision using a novel query-adaptive multi-head attention weighting mechanism. To do that, we first convert box annotations into overlap- and center-aware patch-wise soft labels by combining Intersection over Union (IoU) with the distance labeling according the Gaussian centered at the ground-truth click region (Tang et al., 2025) instead of one-hot patch labeling. This emphasizes the center-click grounding manner in standard GUI protocol (Cheng et al., 2024) while down-weighting partially overlapped border patches. After preparing patch-wise grounding labels, we append a learnable `<ANCHOR>` token after visual and instruction query tokens and supervise its attention to visual tokens as a surrogate aggregator of all query tokens' text-visual attentions. This avoids cumbersome and brittle token-level aggregation in vanilla attention grounding of TAG (Xu et al., 2024a) with mildest interference with the model's pretrained vocabulary. Different from previous work, we first identify visual-sink query tokens that exhibit strong cross-modal alignment with visual inputs by computing similarity using intermediate hidden embeddings. These tokens guide the weighting of attention heads, prioritizing those with robust inter-modal interactions while down-weighting heads that lack meaningful visual-query connections. This approach improves grounding capability without compromising the pretrained capabilities of MLLMs. Together, GUI-AIMA performs visual grounding by manipulating intrinsic attentions and requires no task-specific modules, enhancing its efficiency and generalization.

The core component of GUI-AIMA lies in the head-wise aggregation using the proposed Visual-sink Query Tokens $\mathcal{Q}_s$ for weighting. These tokens supply a query-adaptive prior of the inter-modality patterns of the MLLM, guiding the multi-head weighting according to the uniformity between the inter-modality pattern of each head and the prior pattern found in $\mathcal{Q}_s$. This design preserve the pretrained model's general abilities via respecting the original multi-modal functionality of each attention head, facilitating effective visual grounding training requiring less data.

The main contributions of GUI-AIMA are summarized as follows:

- We introduce GUI-AIMA, an **attention-only**, **coordinate-free** framework that aligns intrinsic MHSA with patch-wise supervision by simplifying the vanilla attention-based visual grounding with anchored-attention predictions and designing an overlap- and center-aware patch-wise labeling scheme that matches human click tendencies better.

- We propose simple and effective visual-sink query tokens and a corresponding multi-head weighting mechanism that emphasizes heads exhibiting strong cross-modal behavior, improving efficiency and generalization without extra grounding modules.

- With only one-stage fine-tuning on a small uncurated open-sourced training sets, GUI-AIMA-3B outperforms prior 3B models and rivaling larger MLLM-based GUI grounding methods. Ablations verify the benefits of the design of anchored attentions, visual-sink head weighting, and weighted patch labels. GUI-AIMA provides insights on how to understand and specialize the functionality of attention heads for visual grounding.

## 2 RELATED WORKS

**Coordinate-based GUI grounding:** The core challenge for GUI agents (Wang et al., 2024b) is grounding: aligning user instructions with the correct actionable elements in the screenshot, due to the semantic inconsistency of layouts and UI elements in diverse GUI environments (Liu et al., 2025). In early attempts, along with the screenshots, extra structured inputs, such as HTML for U-Ground (Gou et al., 2024), or extractions from visual parsing modules, such as element extractions from OmniParser (Wan et al., 2024; Yu et al., 2025) and generated captions as contexts in Aria-UI (Yang et al., 2024), are fed into MLLMs as the supplementation for the better interface

understanding and more precise localization. Later works, such as AGUVIS (Xu et al., 2024b), Seeclick (Cheng et al., 2024) and OS-Atlas (Wu et al., 2024), explore the GUI grounding leveraging MLLMs (Bai et al., 2025; Chen et al., 2024b) with only screenshot inputs, enabling the end-to-end localization with improved scalability on diverse GUI environments. The grounding manner in these methods is to generate coordinate-based click centers or bounding boxes in natural language, which is indirect alignment of visual grounding requiring additional efforts, such as scaled GUI corpora (Qin et al., 2025; Chai et al., 2024) and OCR pretraining (Hong et al., 2024) for connecting coordinates with UI elements. Recent endeavors manage to resolve this gap from the data-intensive manner of the coordinate-based GUI grounding methods via incorporating GUI-specific grounding reward signals in RL-training (Luo et al., 2025; Lu et al., 2025; Liu et al., 2025; Tang et al., 2025) and performing autonomous GUI explorations (Fan et al., 2025; Wu et al., 2025).

**Coordinate-free GUI grounding:** Recent works have begun to replace traditional coordinate-based grounding texts with patch-level attention map predictions, where patches belonging to the correct region receive higher weights. TAG (Xu et al., 2024a) firstly leveraged the multi-head self-attention between GUI instructions and visual tokens in MLLMs for tuning-free GUI grounding, showcasing the intrinsic potential of MLLMs' attention pattern for GUI tasks. However, the generalization of TAG for novel interfaces is restricted by the backbone, and by the heuristics used to select text tokens and attention heads. SE-GUI (Yuan et al., 2025) retains the coordinate-based prediction manner but employs self-attention to iteratively filter out noisy training samples during training. Besides these attention-based improvements, GUI-Actor (Wu et al., 2025) introduces an embedding-based grounding head that derives attention between input visual patch embeddings and the final hidden state of a special <ACTOR> token. Compared with GUI-AIMA, GUI-Actor adds additional module to connect cross-layer visual and <ACTOR> embeddings, requiring extra warm-up training phase and showing much worse training efficiency.

## 3 METHODS

The core idea of GUI-AIMA is utilizing the intrinsic visual-query pattern of MLLMs for GUI grounding. It performs supervised fine-tuning on the multi-head self-attention matrices across layers. Intuitively, as shown in Section 3.1, the patch-wise attention vector between each text token and visual patch tokens can represent which image patches the target element belongs. And the tuning-free aggregation of patch-wise attention vectors between query tokens has shown the potential for text localization (Xu et al., 2024a). Existing barriers for attention-based GUI grounding comes from the aggregation of patch-wise vectors for each query text tokens and each attention head. Besides, annotation gap between patch-wise and original coordinate labels further leads to inaccurate groundings. In GUI-AIMA, we propose a general attention-based GUI grounding framework that simplify the query token selection in Section 3.3 and refine the head-wise weighting mechanism for the patch-wise vector aggregation in Section 3.4. Furthermore, compared with previous embedding-based method, such as GUI-Actor (Wu et al., 2025), GUI-AIMA does not add extra grounding modules to MLLMs, eliminating the extra warm-up training phase.

### 3.1 PRELIMINARY: INHERENT GUI GROUNDING VIA MULTI-HEAD SELF-ATTENTION

**Multi-head Self-Attention (MHSA).** For MLLMs with $L$ layers and $H$ attention heads per layer, given the output embeddings $\mathbf{H}^{l-1}$ of the preceding transformer layer $l-1$ with dimension $d$, query and key are computed as $\mathbf{Q}^{l,h} = \mathbf{H}^{l-1}\mathbf{W}_Q^{l,h}$ and $\mathbf{K}^{l,h} = \mathbf{H}^{l-1}\mathbf{W}_K^{l,h}$ for each self-attention head $h$, $1 \leq h \leq H$ , where $\mathbf{W}_Q^{l,h}, \mathbf{W}_K^{l,h} \in \mathbb{R}^{d \times d_h}$ are parameters with dimension $d_h = d/H$. Then, the self-attention matrix $\mathbf{A}^{l,h}$ is computed as:

$$\mathbf{A}^{l,h} = \text{softmax}\left(\mathbf{Q}^{l,h}\mathbf{K}^{l,h\top}/\sqrt{d_h} + \mathbf{M}\right), \quad \mathbf{M}_{ij} = \begin{cases} -\infty, & i < j \\ 0, & i \geq j \end{cases},$$

where $\mathbf{M}$ is the attention mask for the causal attention $\mathbf{A}^{l,h}$.

**Inherent GUI Grounding Indications in $\mathbf{A}^{l,h}$.** Given the interface image token index sequence $\mathcal{V} = [v_1, \ldots, v_{|\mathcal{V}|}]$ which is patch-wise for MLLMs and the user query token index sequence $\mathcal{Q} =$

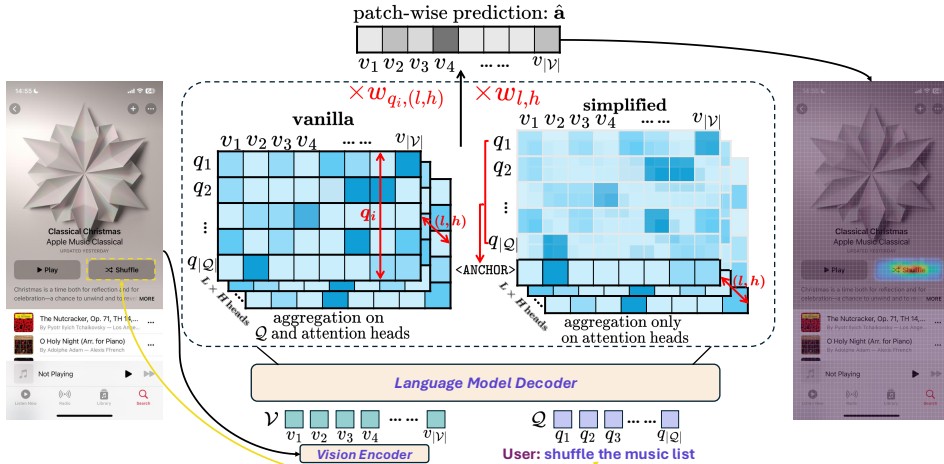

Figure 1: With user instruction sequence $\mathcal{Q}$, screenshot patch sequence $\mathcal{V}$ and multi-head attentions $\{\mathbf{A}^{l,h}\}_{l\in[L],h\in[H]}$ from the MLLM, the vanilla attention grounding in Eq. (1) needs to additionally consider proper aggregation between all query tokens' grounding vectors. In simplified attention grounding of Eq. (4), as <ANCHOR> token learning to be the context anchor of $\mathcal{Q}$, the multi-head aggregation on attention vectorss between <ANCHOR> and $\mathcal{V}$ is adequate for grounding.

$[q_1, \ldots, q_{|\mathcal{Q}|}]$, we format the intermediate token sequence as $\mathbf{H}^{l-1} = [\mathbf{H}_\mathcal{V}^{l-1}, \mathbf{H}_\mathcal{Q}^{l-1}]$ for layer $l-1$, where $\mathbf{H}_\mathcal{V}^{l-1} \in \mathbb{R}^{|\mathcal{V}| \times d}$ is the embedding sequence of visual patch tokens, $\mathbf{H}_\mathcal{Q}^{l-1} \in \mathbb{R}^{|\mathcal{Q}| \times d}$ is the embedding sequence of the user query tokens. With the self-attention matrix $\mathbf{A}^{l,h}$ computed from $\mathbf{H}^{l-1}$, the extracted attention vector $\mathbf{A}_{q_i,\mathcal{V}}^{l,h}$ between each query token $\mathbf{H}_{q_i}^{l-1}$ and the visual sequence $\mathbf{H}_\mathcal{V}^{l-1}$ can reflect the patch-wise query correlation. Among all patches $\mathcal{V}$, the image patch $v_i$ with large attention value $\mathbf{A}_{q_i,v_i}^{l,h}$ is more likely to contain regions related to $q_i$. Intuitively, the aggregation of $\{\mathbf{A}_{q_i,\mathcal{V}}^{l,h}\}_{l\in[L],h\in[H],i\in[|\mathcal{Q}|]}$ among $|\mathcal{Q}|$ text tokens and $L \times H$ attention heads can produce the inherent patch-wise GUI grounding indications:

$$\hat{\mathbf{a}} = \frac{1}{L \cdot H \cdot |\mathcal{Q}|} \sum_{l,h,i} w_{q_i,(l,h)} \, \mathbf{A}_{q_i,\mathcal{V}}^{l,h} \quad \in \mathbb{R}^{|\mathcal{V}|}, \tag{1}$$

where $w_{q_i,(l,h)}$ is the aggregation weights for each query-visual attention vector with $w_{q_i,(l,h)} \geq 0$ and $\sum_{l,h,i} w_{q_i,(l,h)} = 1$. We denote the aggregation in Eq. (1) as the **Vanilla Attention Grounding** shown as "vanilla" in Fig. 1. In the previous training-free attempt (Xu et al., 2024a) based on this manner, every selected text tokens are treated uniformly and attention heads are roughly selected. For the better adaptation of an attention-based GUI grounding method, we need a simplified and effective weighting strategy via $w_{q_i,(l,h)}$.

### 3.2 COORDINATE-FREE PATCH-WISE LABELING

The inherent grounding of MLLMs via the query-visual attention in Eq. (1) is patch-wise instead of the pixel manner in the original coordinate-based bounding boxes. In order to enable the patch-wise grounding supervision, we first reformat the ground truth bounding box $gt^{\text{bbox}} = [x_1, y_1, x_2, y_2]$ into patch-wise vectors $\boldsymbol{p} \in \mathbb{R}^{|\mathcal{V}|}$, where $|\mathcal{V}|$ is the patch size of the interface $I$. Image patch $i$ is positive only if there is an overlap between $gt^{\text{bbox}}$ and patch $i$. Considering the border patches which are partially overlapped with $gt^{\text{bbox}}$ should be less weighted and the patches of center region should be highlighted, we weight each patch with $p_i$ according to the overlapping ratio of itself with $gt^{\text{bbox}}$ and the distance from center of patch $i$ to the center of $gt^{\text{bbox}}$ with Gaussian distribution following GUI-G$^2$ (Tang et al., 2025). In more details, we define the overlap between positive image patch $i$ and grounding box annotation $gt^{\text{bbox}}$ using intersection over union ratio, denoted as $\mathbf{IoU}(i, gt^{\text{bbox}})$. We take its center point $\boldsymbol{\mu}_i = (c_x^i, c_y^i)$ and the center point $\boldsymbol{\mu}_{gt} = (c_x^{gt}, c_y^{gt})$ of $gt^{\text{bbox}}$ for the following weighting scheme:

$$p_{\text{i}} = \mathbf{IoU}(i, gt^{\text{bbox}}) \cdot \mathcal{N}\left(\boldsymbol{\mu}_i; \boldsymbol{\mu}_{gt}, \boldsymbol{\Sigma}_{gt}\right), \tag{2}$$

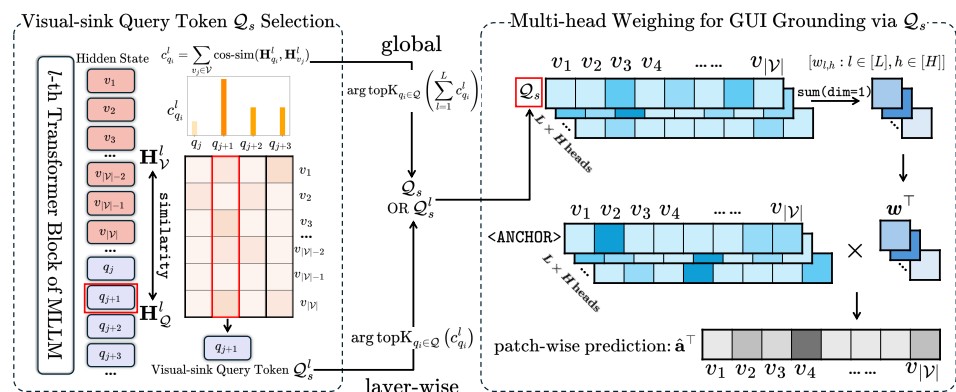

Figure 2: Multi-head attention aggregation strategy for GUI-AIMA: (1) Specify the Visual-sink Query token $\mathcal{Q}_s$ layer-wisely or globally via computing hidden state similarities between query tokens $\mathcal{Q}$ and visual tokens $\mathcal{V}$ in Eq. (7) and Eq. (9); (2) In the case of global selected $\mathcal{Q}_s$, Utilizing $\mathcal{Q}_s$ as the global visual-correlation pattern indicator, GUI-AIMA highlights the attention head in which $\mathcal{Q}_s$ sinks more on visual tokens via Eq. (8).

where $\boldsymbol{\Sigma}_{gt} = \text{diag}\big((\sigma_x^{gt})^2, (\sigma_y^{gt})^2\big)$ with adaptive 2D standard deviations as $\sigma_x^{gt} = \alpha \cdot (x_2 - x_1)$ and $\sigma_y^{gt} = \alpha \cdot (y_2 - y_1)$. Empirically, we set $\alpha = 0.8$. In this manner, coordinate-free weighted patch label $\boldsymbol{p} = \text{normalize}\big(\{p_i\}_{i=1}^M\big)$ is overlapping-aware and encouraging center-clicking.

With the patch-wise prediction $\hat{\mathbf{a}}$ in Eq. (1) and ground truth label $\boldsymbol{p}$ derived from annotations, the attention grounding loss is formulated as the KL-Divergence loss between $\boldsymbol{p}$ and $\hat{\mathbf{a}}$:

$$\mathcal{L}_{\text{GUI\_Attn}} = \mathbb{D}_{\text{KL}}(\boldsymbol{p} \,\|\, \text{normalize}(\hat{\mathbf{a}})) \tag{3}$$

## 3.3 SIMPLIFIED SUPERVISION ON MULTI-HEAD ATTENTIONS WITH VISUAL ANCHOR

The supervision on $\hat{\mathbf{a}}$ in Eq. (1) requires a carefully calibrated balance across $\mathbf{A}_{q_i,\mathcal{V}}^{l,h}$ of each query token, where the weight of each query token $w_{q_i,(l,h)}$ is difficult to determine. Moreover, direct supervision on the attention of query text tokens will impair the MLLM's general capabilities (*e.g.*, image captioning) and thus harm generalization (Nguyen et al., 2023).

To tackle this issue, we consider adding a special <ANCHOR> token into the vocabulary and place it after GUI inputs to format $[\mathcal{V}, \mathcal{Q}, \text{<ANCHOR>}]$. This design simplifies the token-wise aggregation on $\{\mathbf{A}_{q_i,\mathcal{V}}^{l,h}\}_{l\in[L],h\in[H],i\in[|\mathcal{Q}|]}$ and disentangle general understanding functionality with GUI grounding. <ANCHOR> token serves as a bridge between user query tokens and correct grounding patches. Intuitively, for each attention head, the attention vector $\mathbf{A}_{\mathsf{a},\mathcal{V}}^{l,h} \in \mathbb{R}^{|\mathcal{V}|}$, the attention matrices between <ANCHOR> token and all visual tokens $\mathcal{V}$, learns an implicit aggregation of $\{\mathbf{A}_{q_i,\mathcal{V}}^{l,h}\}_{q_i\in\mathcal{Q}}$ over all query tokens. With the anchored attentions, The previous aggregation via the weight $w_{q_i,(l,h)}$ in Eq. (1) can be simplified as:

$$\hat{\mathbf{a}} = \frac{1}{L \cdot H} \sum_{l,h} w_{l,h} \, \mathbf{A}_{\mathsf{a},\mathcal{V}}^{l,h} \quad \in \mathbb{R}^{|\mathcal{V}|}, \tag{4}$$

with the query-ignored weight vector of attention heads $\boldsymbol{w} = [\,w_{l,h} : l \in [L], \; h \in [H]\,] \in \mathbb{R}^{1 \times LH}$. This simplified attention-based grounding is shown as "simplified" in Fig. 1.

## 3.4 MULTI-HEAD WEIGHTING VIA THE MEASUREMENT OF VISUAL-SINK TOKENS

Section 3.3 avoids aggregating attention predictions among all query tokens by introducing a special anchor token. We continue to discuss how to balance $w_{l,h}$, the weights of different heads across different layers. Previous works (Li et al., 2023; Clark et al., 2019) has shown the functional diversity between attention heads across transformer blocks in LLMs. In GUI grounding, we want to identify attention heads, which shows active inter-modality interactions. The attention head with large attention value between visual tokens $\mathcal{V}$ and text query tokens $\mathcal{Q}$ is more relevant, while others are not. As GUI grounding in the attention manner aims to capture the inter-modality correspondence between

the interfaces and user query, the above strategy focusing on multi-modal interactions allows the attention supervision skew towards the inter-modality attention heads and affect less on the neutral attention heads, thus improving the inherent attention grounding of MLLMs without impairing the pretrained capacity, as the adaptation complies the original attention functionalities.

A straightforward measurement of the multi-modality of $\mathbf{A}^{l,h}$ is the cumulative sum of all query-visual attention entries:

$$w_{l,h} = \sum_{q_i,v_i \in \mathcal{Q},\mathcal{V}} \mathbf{A}^{l,h}_{q_i,v_i} \tag{5}$$

This strategy is under-specified as not all text tokens in the query are necessary for inter-modality connections. Since <ANCHOR> is introduced in GUI-AIMA to represent the context of the whole query sequence $\mathcal{Q}$, a similar simplification towards Eq. (5) can be applied as:

$$w_{l,h} = \sum_{v_i \in \mathcal{V}} \mathbf{A}^{l,h}_{\mathtt{a},v_i}, \tag{6}$$

which is the cumulative sum of the attention entry only between visual tokens $\mathcal{V}$ and <ANCHOR> tokens. However, <ANCHOR> is not granted to function as the representative token for context summarization at the initial stage of training, as the embedding of new introduced <ANCHOR> token is random initialized. Only relying on premature <ANCHOR> token as weights in Eq. (6) will bring in noisy and biased attention aggregations. Thus, we consider selecting query-adaptive token for weighting instead of using all query tokens $\{q_i\}$ or <ANCHOR> in Eq. (5) and Eq. (6). We denote the chosen token for computing $w_{l,h}$ as **visual-sink query tokens** $\mathcal{Q}_s$, which are global active tokens for connecting visual inputs and query tokens.

**Visual-sink Query Tokens $\mathcal{Q}_s$ Selection.** For a user query sequence $\mathcal{Q}$, each text token $q_i$ reflects different correlations with visual tokens in $(l, h)$-indexed head as $\sum_{v_i \in \mathcal{V}} \mathbf{A}^{l,h}_{q_i,v_i}$. And the text tokens with the massive visual correlation inside the model vary in different user queries. Here, in order to identify the query tokens with strong visual affinity globally, we directly measure the Cosine similarity with intermediate hidden states $\mathbf{H}^l$ shown as the left part of Fig. 2:

$$c^l_{q_i} = \sum_{v_j \in \mathcal{V}} \mathtt{Sim}(\mathbf{H}^l_{q_i}, \mathbf{H}^l_{v_j}). \tag{7}$$

Tokens with larger $c^l_{q_i}$ has strong visual correlations in layer $l$. The reason of measuring the query-visual affinity of $q_i$ in layer $l$ using hidden states $\mathbf{H}^l$ instead of attention, i.e. $c^l_{q_i} = \sum_{h \in [H]} \sum_{v_i \in \mathcal{V}} \mathbf{A}^{l,h}_{q_i,v_i}$ is that the query-visual pattern discovered in hidden states $\mathbf{H}^l$ is not necessary statistically prevailing among each head's self-attention matrix, as only a smaller subset of attention heads are "semantic heads" that key on semantic functionality and representation similarity reflected in $\mathbf{H}^l$ (Elhelo & Geva, 2024; Olsson et al., 2022; Voita et al., 2019) (an analysis is available in Appendix D). In comparison, the measurement in Eq. (7) is straightforward to reflect visual correlations of each text token in a global view. These "semantic" minorities sharing the similar pattern from $\mathbf{H}$ are attention heads we aim to emphasize. Here, we denote the query tokens with topK large $c^l_{q_i}$ as the **Visual-sink Query Tokens $\mathcal{Q}^l_s$**. As the global "semantic" pattern we extract here is $\mathcal{Q}^l_s$ have large correlations with visual tokens, $\mathcal{Q}^l_s$ in the attention head to be highlighted should be also highly visual-correlated. We compute the cumulative sum of attention values between $\mathcal{Q}^l_s$ and $\mathcal{V}$ in $\mathbf{A}^{l,h}$ for measuring this correlation shown as the right part of Fig. 2, with further normalization:

$$w_{l,h} = \sum_{q^s_i \in \mathcal{Q}^l_s, v_i \in \mathcal{V}} \mathbf{A}^{l,h}_{q^s_i,v_i}, \quad w_{l,h} = \frac{\exp(w_{l,h})}{\sum_{l'=1}^L \sum_{h'=1}^H \exp(w_{l',h'})}. \tag{8}$$

That is, for semantic attention head with the strong query-visual affinity measured as Eq. (8) based on $\mathcal{Q}_s$, it should be emphasized using its large $w_{l,h}$ for aggregations in Eq. (4). In this way, the highlighted attention heads share the aligned global patterns found from the hidden state $\mathbf{H}^l$ as Eq. (7). In practice, we found that unifying the same $\mathcal{Q}_s$ for all layers' $w_{l,h}$, that is $\forall l \in \{1, \ldots, L\}$:

$$\mathcal{Q}^l_s = \mathcal{Q}_s := \underset{q_i \in \mathcal{Q}}{\arg \mathrm{topK}} \left( \sum_{l=1}^L c^l_{q_i} \right), \tag{9}$$

leads to better training stability and performance, instead of layer-wise $\mathcal{Q}_s^l$ different in each layer:

$$\mathcal{Q}_s^l := \arg \operatorname*{topK}_{q_i \in \mathcal{Q}}(c_{q_i}^l). \tag{10}$$

After obtaining $\mathcal{Q}_s$, we compute head-wise weight $w_{l,h}$ via Eq. (8) and perform multi-head aggregation for final prediction $\hat{\mathbf{a}}$ via Eq. (4).

### 3.5 Two-step Inference with Zoom-in

Under GPU memory constraints, high-resolution GUI screenshots are usually down-sampled, yielding fewer visual patch tokens for the MLLM. The resulting information loss and reduced fine-grained spatial granularity inevitably harm grounding accuracy. GUI-AIMA provides flexible spatial granularity since its patch-wise grounding. We can easily perform a two-step inference by adding zoom-in **without extra training**: (1) feed the compressed high-resolution screenshot to predict the approximate location. We use the center of this predicted location to determine a specific area to focus on. (2) we re-run the inference on the newly cropped region, providing a much more accurate result. This two-step inference targets to mitigates failure cases on high-resolution screens where the model identifies the right region but the center prediction is slightly offset the ground-truth box.

## 4 Experiments

**Implementation Details.** We apply Qwen2.5-VL-3B-Instruct (Bai et al., 2025) as the MLLM backbone for GUI-AIMA and all ablation variants. We follow the special token setting in GUI-Actor (Wu et al., 2025) and retain the next-token prediction loss for GUI-AIMA's grounding format. The training of GUI-AIMA-3B is conducted on 8 NVIDIA A100-80G GPUs with effective global batch size 64 and learning rate 5e-6. We set $\alpha$ as 0.8 for the adaptive deviation control in Eq. (2). Regarding to the setting of visual-sink query token $\mathcal{Q}_s$, we select the top-1 global large tokens complying with Eq. (9). As for the training dataset, we employ the entire training set of GUIAct (Chen et al., 2024a), AndroidControl (Li et al., 2024), Wave-UI (Jeffries & Team, 2024), randomly sampled 60k samples from UGround (Gou et al., 2024) and from GTA1 training set (Yang et al., 2025), respectively, with 254k instructions and 85k images in all. For the 45k ablation training set for Section 4.2, we select the first 10k samples from GUIAct, AndroidControl, Wave-UI and first 15k samples from UGround. GUI-AIMA-3B is trained for one epoch with all parameters unfrozen, without extra modules and a warmup stage. Implementation of baselines is in Appendix E.

**Efficient Extraction of Multi-head Self-Attention** Fast and memory-efficient attention implementations, such as FlashAttention Dao et al. (2022), avoid materializing the full attention matrix and therefore do not store intermediate attention weights. By contrast, the eager execution in original transformer attention Vaswani et al. (2017) can return the full attention maps, but it is neither fast nor memory-efficient, which becomes a bottleneck for large-scale training and for scaling up model size. In GUI-AIMA, we address this by combining FlashAttention with eager attention: we use FlashAttention for the regular layer-to-layer forward pass, and then reuse the same attention parameters to compute a partial attention map, covering the rows of text tokens and the `<ANCHOR>` token, via eager execution for GUI-AIMA's attention grounding. Since visual tokens dominate the attention in MLLMs, the additional computation cost is slight as shown in Table 8 of Appendix B.

**Evaluation datasets and metrics.** We evaluate GUI-AIMA, baselines and ablation variants on ScreenSpot-v2 (Wu et al., 2024) for general GUI visual grounding evaluation across mobile, desktop and web domains, ScreenSpot-Pro (Li et al., 2025a) for testing on challenging higher-resolution screenshots from diverse and complex professional software scenarios, e.g. interfaces from different Operating Systems with large distribution gap, and OSWorld-G Xie et al. (2025). Both benchmarks contain separated text-centered and icon-centered visual grounding tasks, where the icon set is the more abstract visual-based task less hinted by text. For grounding accuracy, we follow the standard center point-based metric (Lin et al., 2024) that the correct click-point prediction locates within the ground-truth bounding box.

**Baselines.** 3 categories of methods are compared: (1) General MLLMs including GPT-4o (OpenAI, 2024), Claude Computer (Hu et al., 2024), Operator (OpenAI, 2025), Qwen2.5-VL (Bai et al.,

Table 1: Performance comparison of different models across various task categories based on Text, Icon, and Average scores on **ScreenSpot-Pro**. "-" indicates unreported results in original papers. Methods with $*$ are Qwen-2.5-VL-based. $\dagger$ denotes the training-free method.

| Model | CAD Text | CAD Icon | Dev Text | Dev Icon | Creative Text | Creative Icon | Scientific Text | Scientific Icon | Office Text | Office Icon | OS Text | OS Icon | Avg. Text | Avg. Icon | Avg. |
|---|---|---|---|---|---|---|---|---|---|---|---|---|---|---|---|
| **General** | | | | | | | | | | | | | | | |
| GPT-4o | 2.0 | 0.0 | 1.3 | 0.0 | 1.0 | 0.0 | 2.1 | 0.0 | 1.1 | 0.0 | 0.0 | 0.0 | 1.3 | 0.0 | 0.8 |
| Claude Computer | 14.5 | 3.7 | 22.0 | 3.9 | 25.9 | 3.4 | 33.9 | 15.8 | 30.1 | 16.3 | 11.0 | 4.5 | 23.4 | 7.1 | 17.1 |
| Qwen2.5-VL-3B | 9.1 | 7.3 | 22.1 | 1.4 | 26.8 | 2.1 | 38.2 | 7.3 | 33.9 | 15.1 | 10.3 | 1.1 | 23.6 | 3.8 | 16.1 |
| Qwen2.5-VL-7B | 16.8 | 1.6 | 46.8 | 4.1 | 35.9 | 7.7 | 49.3 | 7.3 | 52.5 | 20.8 | 37.4 | 6.7 | 38.9 | 7.1 | 26.8 |
| **SFT** | | | | | | | | | | | | | | | |
| OS-Atlas-7B | 12.2 | 4.7 | 33.1 | 1.4 | 28.8 | 2.8 | 37.5 | 7.3 | 33.9 | 5.7 | 27.1 | 4.5 | 28.1 | 4.0 | 18.9 |
| UGround-V1-7B | 15.8 | 1.2 | 51.9 | 2.8 | 47.5 | 9.7 | 57.6 | 14.5 | 60.5 | 13.2 | 38.3 | 7.9 | 45.2 | 8.1 | 31.1 |
| UI-TARS-72B | 18.8 | 12.5 | 62.9 | 17.2 | 57.1 | 15.4 | 64.6 | 20.9 | 63.3 | 26.4 | 42.1 | 15.7 | 50.9 | 17.6 | 38.1 |
| JEDI-3B$*$ | 27.4 | 9.4 | 61.0 | 13.8 | 53.5 | 8.4 | 54.2 | 18.2 | 64.4 | 32.1 | 38.3 | 9.0 | 49.8 | 13.7 | 36.1 |
| JEDI-7B$*$ | 38.0 | 14.1 | 42.9 | 11.0 | 50.0 | 11.9 | **72.9** | 25.5 | **75.1** | **47.2** | 33.6 | 16.9 | 52.6 | 18.2 | 39.5 |
| UI-TARS-1.5-7B | 38.6 | 11.0 | 58.4 | 12.4 | **58.1** | 15.4 | 66.7 | 21.9 | 74.6 | 35.9 | **49.5** | 13.5 | 57.5 | 16.9 | 42.0 |
| GMS$\dagger$ (Qwen2.5-VL-7B) | 59.4 | 29.7 | 53.2 | 20.7 | 57.1 | 19.6 | 62.5 | 34.5 | 67.8 | 35.8 | 45.8 | 15.7 | 58.4 | 24.5 | 45.5 |
| GUI-Actor-3B$*$ | - | - | - | - | - | - | - | - | - | - | - | - | - | - | 42.2 |
| GUI-Actor-3B$*$ + Verifier | - | - | - | - | - | - | - | - | - | - | - | - | - | - | 45.9 |
| GUI-Actor-7B$*$ | - | - | - | - | - | - | - | - | - | - | - | - | - | - | 44.6 |
| GUI-Actor-7B$*$ + Verifier | - | - | - | - | - | - | - | - | - | - | - | - | - | - | 47.7 |
| **RL** | | | | | | | | | | | | | | | |
| UI-R1-E-3B$*$ | 37.1 | 12.5 | 46.1 | 6.9 | 41.9 | 4.2 | 56.9 | 21.8 | 65.0 | 26.4 | 32.7 | 10.1 | - | - | 33.5 |
| InfiGUI-R1-3B$*$ | 33.0 | 14.1 | 51.3 | 12.4 | 44.9 | 7.0 | 58.3 | 20.0 | 65.5 | 28.3 | 43.9 | 12.4 | 49.1 | 14.1 | 35.7 |
| GUI-G1-3B$*$ | 39.6 | 9.4 | 50.7 | 10.3 | 36.6 | 11.9 | 61.8 | 30.0 | 67.2 | 32.1 | 23.5 | 10.6 | 49.5 | 16.8 | 37.1 |
| SE-GUI-3B$*$ | 38.1 | 12.5 | 55.8 | 7.6 | 47.0 | 4.9 | 61.8 | 16.4 | 59.9 | 24.5 | 40.2 | 12.4 | 50.4 | 11.8 | 35.9 |
| GUI-G$^2$-3B$*$ | 20.3 | 9.4 | 42.2 | 2.1 | 43.4 | 2.8 | 43.8 | 10.0 | 46.3 | 15.1 | 29.0 | 4.5 | 37.6 | 6.0 | 25.5 |
| **GUI-AIMA-3B$*$** | 44.7 | 21.9 | 68.2 | 28.3 | 59.6 | 21.7 | 69.4 | 37.3 | 70.6 | 49.1 | 65.4 | 31.5 | 62.0 | 30.0 | 49.8 |
| **GUI-AIMA-3B$*$ + zoom-in** | 54.3 | 26.6 | **77.9** | **37.2** | 67.2 | 31.5 | 76.4 | 39.1 | 80.8 | 64.2 | 61.7 | 32.6 | **69.5** | **36.8** | **57.0** |

2025); (2) GUI-specific supervised fine-tuned model: OS-Atlas (Wu et al., 2024), UGround (Gou et al., 2024), SparkUI-Parser Jing et al. (2025), UI-TARS (Qin et al., 2025), JEDI (Xie et al., 2025) and GUI-Actor (Wu et al., 2025) ; (3) GUI-specific Reinforcement fine-tuned model: UI-R1 (Lu et al., 2025), InfiGUI-R1 (Liu et al., 2025), GUI-G1 (Zhou et al., 2025), SE-GUI (Yuan et al., 2025), GUI-G$^2$ (Tang et al., 2025); (4) Training-free multi-model method: GMS Li et al. (2025b).

## 4.1 MAIN RESULTS: GROUNDING PERFORMANCE

In Table 1, Table 2 and Table 3, we compare GUI-AIMA with state-of-the-art methods on ScreenSpot-Pro, ScreenSpot-v2 and OSWorld-G. While only being trained with 254k samples without filtering or careful selection, GUI-AIMA-3B outperforms all same scale 3B MLLM-based models and shows comparable or even better results than larger scaled models. Among Qwen-2.5-VL based models, marked with $*$ in both tables, GUI-AIMA-3B can better handle the visual grounding in complex software scenarios across different platforms and achieves the best performance on the most challenging ScreenSpot-Pro benchmark, especially on the abstract icon task set. Specifically, GUI-AIMA-3B is better than strong large size coordinate-based UI-TARS-1.5-7B, JEDI-7B, and also better than the embedding-based coordinate-free GUI-Actor-7B model, highlighting the superiority of directly supervising on the multi-head self-attention weights instead of modeling the query-visual attention map via hidden states. On ScreenSpot-v2, GUI-AIMA-3B achieve the comparable results with the strongest baselines, such as JEDI-7B and UI-TARS-7B, and better than the same size GUI-Actor-3B. Besides the advanced performance, another advantage is the training efficiency of GUI-AIMA with only 254k training elements, as most supervised fine-tuned baselines trained on millions of GUI elements. For the comparison with reinforcement fine-tuned baselines, while both trained with smaller training sets than SFT baselines, GUI-AIMA-3B performs better and shows better generalization. The two-step inference with zoom-in without extra training significantly improves the performance of GUI-AIMA on high-resolution benchmarks, ScreenSpot-pro and OSWorld-G, specifically 57.0% on ScreenSpot-pro and 63.8% on OSWorld-G using GUI-AIMA-3B, demonstrating the flexibility of GUI-AIMA's attention-based patch-wise grounding for inference-time improvements.

Table 2: Performance comparison of different models across various task categories based on Mobile, Desktop, Web and Average scores on **ScreenSpot-v2**. "-" indicates unreported results in original papers. Methods with $*$ are Qwen-2.5-VL-based.

| | Model | Mobile | | Desktop | | Web | | Avg. |
|---|---|---|---|---|---|---|---|---|
| | | Text | Icon | Text | Icon | Text | Icon | |
| General | Operator | 47.3 | 41.5 | 90.2 | 80.3 | 92.8 | 84.3 | 70.5 |
| | GPT-4o + OmniParser-v2 | 95.5 | 74.6 | 92.3 | 60.9 | 88.0 | 59.6 | 80.7 |
| | Qwen2.5-VL-3B | 93.4 | 73.5 | 88.1 | 58.6 | 88.0 | 71.4 | 80.9 |
| | Qwen2.5-VL-7B | 97.6 | 87.2 | 90.2 | 74.2 | 93.2 | 81.3 | 88.8 |
| SFT | OS-Atlas-7B | 95.2 | 75.8 | 90.7 | 63.6 | 90.6 | 77.3 | 84.1 |
| | UGround-V1-7B | 95.0 | 83.3 | 95.0 | 77.8 | 92.1 | 77.2 | 87.6 |
| | SparkUI-Parser | 96.9 | 87.7 | 95.9 | 77.1 | 93.4 | 86.2 | 89.5 |
| | UI-TARS-1.5-7B | 95.9 | 84.8 | 94.9 | 80.7 | 90.6 | 86.2 | 89.7 |
| | UI-TARS-7B | 96.9 | 89.1 | 95.4 | 85.0 | 93.6 | 85.2 | 91.6 |
| | JEDI-3B$^*$ | 96.6 | 81.5 | **96.9** | 78.6 | 88.5 | 83.7 | 88.6 |
| | JEDI-7B$^*$ | 96.9 | 87.2 | 95.9 | 87.9 | 94.4 | 84.2 | 91.7 |
| | GUI-Actor-3B | 97.6 | 83.4 | **96.9** | 83.6 | 94.0 | 85.7 | 91.0 |
| | GUI-Actor-3B + Verifier | 98.3 | 85.3 | **96.9** | 87.9 | 95.3 | 86.7 | **92.4** |
| RL | UI-R1-E-3B$^*$ | 83.0 | **97.1** | 85.0 | **91.7** | 77.9 | **95.4** | 89.2 |
| | GUI-G$^2$-3B$^*$ | 96.5 | 82.5 | 95.4 | 75.0 | 88.4 | 72.4 | 86.3 |
| | GUI-AIMA-3B | **99.2** | 85.9 | 96.1 | 88.9 | **96.1** | 80.2 | 91.5 |

Table 3: Performance comparison of different models across various task categories based on Mobile, Desktop, Web and Average scores on **OSWorld-G**. Methods with $*$ are Qwen-2.5-VL-based.

| | Model | Text Matching | Element Recognition | Layout Understanding | Fine-grained Manipulation | Avg. |
|---|---|---|---|---|---|---|
| General | Operator | 51.3 | 42.4 | 46.6 | 31.5 | 40.6 |
| | Gemini-2.5-Pro | 59.8 | 45.5 | 49.0 | 33.6 | 45.2 |
| | Qwen2.5-VL-3B | 41.4 | 28.8 | 34.8 | 13.4 | 27.3 |
| | Qwen2.5-VL-7B | 45.6 | 32.7 | 41.9 | 18.1 | 31.4 |
| SFT | OS-Atlas-7B | 44.1 | 29.4 | 35.2 | 16.8 | 27.7 |
| | UGround-V1-7B | 51.3 | 40.3 | 43.5 | 24.8 | 36.4 |
| | UI-TARS-7B | 60.2 | 51.8 | 54.9 | 35.6 | 47.5 |
| | JEDI-7B$^*$ | 65.9 | 55.5 | 57.7 | 46.9 | 54.1 |
| | GUI-Actor-7B$^*$ | 65.9 | 62.7 | 66.4 | 38.2 | 56.6 |
| | UI-TARS-1.5-7B | 52.6 | 75.4 | 72.4 | 66.7 | **64.2** |
| | JEDI-3B$^*$ | **67.4** | 53.0 | 53.8 | 44.3 | 50.9 |
| | GUI-Actor-3B$^*$ | 64.4 | 60.6 | 64.8 | 33.6 | 54.6 |
| | **GUI-AIMA-3B**$^*$ | 64.8 | 65.5 | 68.8 | 36.8 | 58.3 |
| | **GUI-AIMA-3B**$^*$+ zoom-in | **71.3** | **70.6** | **73.1** | **47.4** | **63.8** |

## 4.2 Ablations

Table 4 shows the ablation results on ScreenSpot-v2 and ScreenSpot-pro. All the ablation variants are trained on the 45k ablation training data with Qwen2.5-VL-3B-Instruct as the backbone. GUI-Actor (45k) is trained with two stages same as the original setting.

**Compare Coordinate-free modeling manners.** In Table 4, we include 3 different coordinate-free GUI grounding manners, one embedding-based: GUI-Actor which relies on hidden embeddings for computing similarity with extra modules, and two attention-based: vanilla attention grounding as Eq. (1) and simplified attention grounding as Eq. (4). With the same importance on all the query tokens and attention heads, the vanilla attention grounding can converge faster than GUI-Actor on more complex visual grounding tasks in ScreenSpot-Pro. When simplifying the aggregation on query tokens, even with naive "weighting uniformly", the simplified attention grounding can have **1.50%** and **4.24%** improvement on GUI-Actor for ScreenSpot-v2 and ScreenSpot-Pro. These results demonstrate the effectiveness and faster convergence of attention-based visual grounding methods than embedding-based methods, and shows the advantage of compressing query contexts into <ANCHOR> token instead of manually control the grounding importance from each query token.

Table 4: Ablation results on ScreenSpot-v2 and ScreenSpot-Pro of coordinate-free GUI grounding methods fine-tuned with a fixed 45k randomly sampled dataset. Variants in blue represent the selected settings for GUI-AIMA.

| Model | ScreenSpot-v2↑ | | | ScreenSpot-Pro↑ | | |
|---|---|---|---|---|---|---|
| | Text | Icon | Avg. | Text | Icon | Avg. |
| *Existing Coordinate-free GUI Grounding* | | | | | | |
| GUI-Actor-3B (45k) | 96.24 | 75.99 | 87.42 | 50.67 | 12.25 | 35.99 |
| Vanilla Attention Grounding (45k) | 95.68 | 75.45 | 86.87 | 53.02 | 12.42 | 37.51 |
| | | | | | | |
| *Simplified Attention Grounding: GUI-Attention-3B (45k)* | | | | | | |
| *Multi-head Weighting without $\mathcal{Q}_s$* | | | | | | |
| + weighting uniformly | 97.08 | 78.34 | 88.92 | 56.50 | 13.91 | 40.23 |
| + weighting with all $\mathcal{Q}$ Eq. (5) | 96.24 | 78.34 | 88.44 | 52.61 | 13.41 | 37.63 |
| + weighting with <ANCHOR> in Eq. (6) | 96.66 | 76.35 | 87.81 | 56.91 | 14.57 | 40.73 |
| *Multi-head weighting with $\mathcal{Q}_s$ in Eq. (8)* | | | | | | |
| + layer-wise top-1 $\mathcal{Q}_s^l$ | 96.52 | **81.23** | **89.86** | 57.32 | **15.73** | 41.43 |
| + layer-wise top-3 $\mathcal{Q}_s^l$ | 96.66 | 79.06 | 88.99 | 57.83 | 15.07 | 41.49 |
| + global top-1 $\mathcal{Q}_s$ | **97.49** | 78.88 | 89.39 | 59.26 | 14.40 | 42.13 |
| + global top-3 $\mathcal{Q}_s$ | 95.82 | 70.94 | 84.98 | 57.01 | 14.40 | 40.73 |
| + weighted patch-wise labeling | 97.21 | 79.60 | 89.54 | **61.00** | 14.90 | **43.39** |

Figure 3: Convergence on ScreenSpot-Pro of GUI-AIMA, GUI-Actor and vanilla attention grounding trained on 45k ablation dataset.

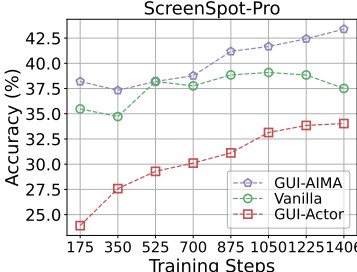

**GUI grounding without Visual-sink Query Token $\mathcal{Q}_s$.** For attention-based GUI grounding variants, weighting the query-visual correlation of each head via the cumulative sum of all query's predictions in Eq. (5) or only the prediction from <ANCHOR> in Eq. (6) leads to more biased attention predictionss than uniform weighting, with worse ScreenSpot-v2 performance of both variants and no improvement on ScreenSpot-Pro as shown in "Multi-head Weighting without $\mathcal{Q}_s$" part of Table 4. This bias comes from the misleading weighting from visual-unrelated tokens for the variants using all $\mathcal{Q}$ and from using the premature <ANCHOR> token for grounding in the later variant.

**The effect of Visual-sink Query Token $\mathcal{Q}_s$.** For ablation of using $\mathcal{Q}_s$ for $w_{l,h}$, we vary the K value between 1 and 3 in the topK selection of $\mathcal{Q}_s$ tokens and explore on whether differing $\mathcal{Q}_s^l$ for each MLLM's layer as in Eq. (10) or using the same $\mathcal{Q}_s$ for all layers as in Eq. (9). In Table 4, we can observe that the layer-wise manner performs slightly better on the icon-based tasks, but global-selected uniform $\mathcal{Q}_s$ leads to more balanced results on both text and icon sets. Among both manners, top-1 selection remains the overall best performance. These ablations verify the "global top-1 $\mathcal{Q}_s$" setting of GUI-AIMA, with **1.90%** improvement over "weighting uniformly" on ScreenSpot-Pro.

**Patch-wise labeling considering overlapping and distance.** Here, we also justify the overlapping- and distance-aware (Tang et al., 2025) patch-wise labeling for fine-tuning GUI-AIMA in Eq. (2). Based on "global top-1 $\mathcal{Q}_s$", down-weighting partial positive and distanced image patch instead of labeling all patch equally leads to **1.26%** further enhancement on ScreenSpot-Pro.

**Training efficiency.** In Fig. 3 , we show the convergence tendency of three methods: GUI-AIMA, vanilla attention grounding and GUI-Actor (with extra warm-up stage already) initialized from Qwen-2.5-VL-3B. GUI-AIMA converges fastest and achieves the best final results, with steady improvements. Vanilla attention grounding fluctuates as supervision on all tokens from the pretrained vocabulary impairs the general capacity. And GUI-Actor needs longer training period to customize its extra modules for GUI grounding.

## 5 CONCLUSION

We presented GUI-AIMA, an attention-only, coordinate-free approach to GUI visual grounding that aligns intrinsic multi-head self-attention with patch-wise supervision. With soft, overlap- and center-aware patch labels converted from coordinate-based annotations, GUI-AIMA simplifies the vanilla attention visual grounding via a learnable <ANCHOR> token as the surrogate to aggregate query-to-visual attention heads. To effectively aggregate different attention heads, we first identify visual-sink query tokens based on cross-modal similarity of hidden states. Experimental results of GUI-AIMA on ScreenSpot-v2 and the challenging ScreenSpot-Pro benchmarks, showing state-of-the-art performance at the 3B scale and rivals other methods with larger MLLM backbones, using only about 85k training screenshots. Ablations further verify the effectiveness of each design choice, such as anchored attention aggregation, query-adaptive head weighting with visual-sink query tokens, and weighted patch labels. For the future works, extending GUI-AIMA to more general and complex visual grounding tasks are pending explorations.

## 6 ETHICS STATEMENT

The proposed GUI-AIMA focuses on MLLMs for the graphical user interface (GUI) grounding for general assistant purpose. All experiments are conducted using publicly available datasets that contain synthetic or anonymized user interface screenshots, charts, and documents. No personally identifiable information or sensitive user data is included.

## 7 REPRODUCIBILITY STATEMENT

The implementation details of proposed method and baselines are detailed list in Section 4 and Appendices E and F. Detailed ablations in 4.2 and visualization examples in G further support the implementations. All training and testing data are open-source dataset, with the detailed training recipe introduced in Section 4.

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

Table 5: Performance comparison of different models across various task categories based on Mobile, Desktop, Web and Average scores on **ScreenSpot-v1**.

| | Model | Mobile | | Desktop | | Web | | Avg. |
|---|---|---|---|---|---|---|---|---|
| | | Text | Icon | Text | Icon | Text | Icon | |
| | GPT-4 | 22.6 | 24.5 | 20.2 | 11.8 | 9.2 | 8.8 | 16.2 |
| | GPT-4o | 20.2 | 24.9 | 21.1 | 23.6 | 12.2 | 7.8 | 18.3 |
| | Claude Computer Use | - | - | - | - | - | - | 83.0 |
| | Gemini 2.0 | - | - | - | - | - | - | 84.0 |
| **3B** | UGround-V1-2B | 89.4 | 72.0 | 88.7 | 65.7 | 81.3 | 68.9 | 77.7 |
| | UI-TARS-2B | 93.0 | 75.5 | 90.7 | 68.6 | 84.3 | 74.8 | 82.3 |
| | GUI-Actor-2B (Qwen2-VL) | 93.0 | 79.9 | 88.1 | 78.6 | 90.9 | 84.0 | 86.5 |
| | GUI-Actor-3B (Qwen2.5-VL) | 94.5 | 83.8 | 92.8 | 82.1 | 91.3 | 82.5 | 88.4 |
| | **GUI-AIMA-3B** | 97.1 | 82.1 | 95.4 | 83.6 | 90.9 | 80.6 | **88.8** |
| **7B** | Qwen2-VL-7B | 75.5 | 60.7 | 76.3 | 54.3 | 35.2 | 25.7 | 55.3 |
| | CogAgent-7B | 67.0 | 24.0 | 74.2 | 20.0 | 70.4 | 28.6 | 47.4 |
| | SeeClick-9.6B | 78.0 | 52.0 | 72.2 | 30.0 | 55.7 | 32.5 | 53.4 |
| | Magma-8B | 60.4 | 58.5 | 75.3 | 52.9 | 69.1 | 52.0 | 60.3 |
| | Aguvis-G-7B | 88.3 | 78.2 | 88.1 | 70.7 | 85.7 | 74.8 | 81.8 |
| | OS-Atlas-7B | 93.0 | 72.9 | 91.8 | 62.9 | 90.9 | 74.3 | 82.5 |
| | Aguvis-7B | 95.6 | 77.7 | 93.8 | 67.1 | 88.3 | 75.2 | 84.4 |
| | UGround-v1-7B | 93.0 | 79.9 | 93.8 | 76.4 | 90.9 | 84.0 | 86.3 |
| | SparkUI-Parser | 94.9 | 83.8 | 95.9 | 80.7 | 89.6 | 82.9 | 88.0 |
| | UI-TARS-7B | 94.5 | 85.2 | 95.9 | 85.7 | 90.0 | 83.5 | 89.5 |
| | GUI-Actor-7B (Qwen2-VL) | 94.9 | 82.1 | 91.8 | 80.0 | 91.3 | 85.4 | 88.3 |
| | GUI-Actor-7B (Qwen2.5-VL) | 96.3 | 85.2 | 95.4 | 82.9 | 90.4 | 85.4 | **89.9** |
| **72B** | UI-TARS-72B | 94.9 | 82.5 | 89.7 | 88.6 | 88.7 | 85.0 | 88.4 |
| | Aguvis-72B | 94.5 | 85.2 | 95.4 | 77.9 | 91.3 | 85.9 | 89.2 |
| | UGround-V1-72B | 94.1 | 83.4 | 94.9 | 85.7 | 90.4 | 87.9 | **89.4** |

Table 6: Performance of GUI-AIMA and GUI-Actor Wu et al. (2025) using the grounding verifier of GUI-Actor on ScreenSpot-v1 and ScreenSpot-v2.

| Model | Mobile | | Desktop | | Web | | Avg. |
|---|---|---|---|---|---|---|---|
| | Text | Icon | Text | Icon | Text | Icon | |
| GUI-Actor-3B (Qwen2.5-VL) + Verifier | 95.6 | 84.7 | 93.8 | 83.6 | 93.0 | 83.0 | 89.5$_{(+1.1)}$ |
| GUI-Actor-7B (Qwen2.5-VL) + Verifier | 96.0 | 85.6 | 96.4 | 82.9 | 91.7 | 83.0 | **89.9**$_{(+0.0)}$ |
| **GUI-AIMA-3B** + Verifier | 97.1 | 83.8 | 96.4 | 83.6 | 91.3 | 84.0 | **89.9**$_{(+1.1)}$ |
| GUI-Actor-3B (Qwen2.5-VL) + Verifier | 98.3 | 85.3 | 96.9 | 87.9 | 95.3 | 86.7 | 92.4$_{(+1.4)}$ |
| GUI-Actor-7B (Qwen2.5-VL) + Verifier | 96.9 | 89.6 | 97.4 | 86.4 | 95.7 | 84.7 | 92.5$_{(+0.4)}$ |
| **GUI-AIMA-3B** + Verifier | 99.0 | 87.2 | 99.0 | 89.3 | 96.6 | 83.3 | **93.0**$_{(+1.5)}$ |

## A  THE USE OF LARGE LANGUAGE MODELS (LLMS)

We used large language models as a general-purpose assist tool limited for correcting grammar and typos and making minor stylistic edits to author-written text. It was not used as any other roles.

## B  EXTRA RESULTS

In Table 5, we show the result of GUI-AIMA-3B and baselines on ScreenSpot-v1 Cheng et al. (2024). In Table 6, we compare the performance of GUI-AIMA and GUI-Actor, with both combined with the multi-region verifier proposed by GUI-Actor. In Table 8, we compare the inference and training time between GUI-AIMA-3B and GUI-Actor-3B.

Table 7: Ablation results on ScreenSpot-v2, ScreenSpot-Pro and OSWorld-G of GUI grounding methods fine-tuned with the entire training set (254k elements). All variants are trained with the weighted patch-wise label introduced in Section 3.2 except *.

| Model | ScreenSpot-v2 | | | ScreenSpot-Pro | | | OSWorld-G |
|---|---|---|---|---|---|---|---|
| | Text | Icon | Avg. | Text | Icon | Avg. | Avg. |
| GUI-Actor-3B* w/o weighted patch-wise label | 94.99 | 80.87 | 88.84 | 49.13 | 19.04 | 37.63 | 50.53 |
| GUI-Actor-3B | 95.13 | 83.94 | 90.25 | 54.04 | 24.50 | 42.76 | 55.14 |
| *Attention Grounding without* <ANCHOR> | | | | | | | |
| Vanilla Attention Grounding (multi-head weighting uniformly) | 95.40 | 81.41 | 89.31 | 58.03 | 21.52 | 44.09 | 55.03 |
| Vanilla Attention Grounding (multi-head weighted by Eq. (5)) | 95.40 | 81.23 | 89.23 | 55.07 | 23.84 | 43.14 | 54.14 |
| *Simplified Attention Grounding with* <ANCHOR> | | | | | | | |
| <ANCHOR>-based multi-head weighting in Eq. (6) | 96.52 | 82.85 | 90.57 | 59.77 | 21.69 | 45.22 | 54.79 |
| GUI-AIMA-3B w/ stop-grad on $w$ | 96.38 | 85.56 | **91.67** | 58.34 | 25.33 | 45.73 | 56.38 |
| GUI-AIMA-3B | 97.21 | 84.12 | 91.51 | 62.03 | 29.97 | **49.78** | **58.33** |

Table 8: Inference and training time comparison between GUI-AIMA-3B and GUI-Actor-3B.

| | GUI-Actor-3B | GUI-AIMA-3B |
|---|---|---|
| ScreenSpot-v2 (s/iteration) | 0.73 | 0.76 |
| ScreenSpot-Pro (s/iteration) | 1.72 | 1.76 |
| OSWorld-G (s/iteration) | 1.15 | 1.23 |
| Training time (A100 GPU-hours) | $\sim 260$ | $\sim 192$ |

## C  FUTURE WORK: EXTENDING GUI-AIMA TO MULTI-ACTION AND MULTI-ELEMENT GROUNDING

**Current limitation.**   GUI-AIMA currently is a grounding-only method for a single grounding element. It utilizes a single learnable <ANCHOR> token to explicitly model single-element grounding, but it neither represents the action type (e.g., click, drag-and-pull) nor supports multi-element actions that require multiple grounding predictions in a single iteration, which limits its applicability as a full GUI agent. Therefore, we propose to extend GUI-AIMA as a full-functional GUI agent that also works for multi-element actions.

**Classification token for navigation.**   To extend GUI-AIMA for multi-action navigation, we can introduce an additional <CLS> token before grounding tokens and format the input as

$$[\mathcal{V}, \mathcal{Q}, \texttt{<CLS>}, \texttt{<ANCHOR\_1>}, \dots, \texttt{<ANCHOR\_M>}],$$

where $M$ is the maximum number of grounded arguments allowed for an action. Let $\mathbf{h}_{\text{CLS}}^L \in \mathbb{R}^d$ denote the final-layer hidden state of <CLS>. We add a classification head over $\mathbf{h}_{\text{CLS}}^L$ and, in the example of a single linear layer, let $u = \mathbf{W}_{\text{CLS}} \cdot \mathbf{h}_{\text{CLS}}^L \in \mathbb{R}^{|\mathcal{A}|}$ be the logits over the discrete action space $\mathcal{A}$ (e.g., click, drag-and-pull, scroll). Given a ground-truth action label $y \in \{1, \dots, |\mathcal{A}|\}$, we define a standard classification loss for each action selection:

$$\mathcal{L}_{\text{act}} = -\log \frac{\exp(u_y)}{\sum_{c=1}^{|\mathcal{A}|} \exp(u_c)} , \tag{11}$$

which encourages the <CLS> token to capture high-level navigation decisions conditioned on the screenshot and instruction.

**Conditioned number of grounding tokens.**   Importantly, the number of grounding tokens is not fixed globally, but can be conditioned on the predicted action type. For each action label $y$, we define $M_y$ as the number of required grounded arguments: for instance, a click action only needs $M_y = 1$ anchor, whereas a drag-and-pull action can be represented with $M_y = 2$ anchors (source and target). At training time, once the ground-truth action $y$ is known, we only supervise the first $M_y$ anchor tokens

$$\texttt{<ANCHOR\_1>}, \dots, \texttt{<ANCHOR\_M}_y\texttt{>},$$

and ignore the remaining anchors (if any). At inference time, the model first predicts the action type via <CLS> token, and then allocates the corresponding number of anchor tokens according to $M_y$, enabling on-demand grounding for different actions.

**Multi-element actions and anchor–element binding.** For multi-element actions, each action element is explicitly bound to a specific anchor token during training. Let $\hat{\mathbf{a}}^{(k)} \in \mathbb{R}^{|V|}$ denote the patch-wise prediction generated in the same manner of GUI-AIMA. For the $k$-th argument of action $y$, we construct a patch-wise label $\boldsymbol{p}^{(k)} \in \mathbb{R}^{|V|}$ corresponding to the $k$-th grounded element. We then define the multi-anchor attention loss as

$$\mathcal{L}_{\text{GUI\_Attn}} = \frac{1}{M_y} \sum_{k=1}^{M_y} \text{KL}\Big(\boldsymbol{p}^{(k)} \,\|\, \text{normalize}\big(\hat{\mathbf{a}}^{(k)}\big)\Big), \tag{12}$$

which reduces to the original single-anchor loss when $M_y = 1$. A concrete example is a drag-and-pull action: we set $M_y = 2$ and bind `<ANCHOR_1>` to the initial region (source element) for dragging and `<ANCHOR_2>` to the end region (target element) after pulling. During training, this binding is fixed by the ordering of ground-truth elements, so that each anchor consistently learns one semantic role of the action.

**Action-specific Grounding Token (optional).** While the above formulation assumes a shared `<ANCHOR_k>` vocabulary across action types, another option is to specialize different grounding anchor tokens for different actions, e.g., `<ANCHOR_DRAG_SRC>` and `<ANCHOR_DRAG_DST>` for drag-and-pull.

We leave implementation of this extension (enabling grounding and planning, for single- and multi-element tasks) as future work, since the current paper focuses on how to make the attention grounding manner effective.

# D ANALYSIS FOR VISUAL-SINK QUERY TOKEN

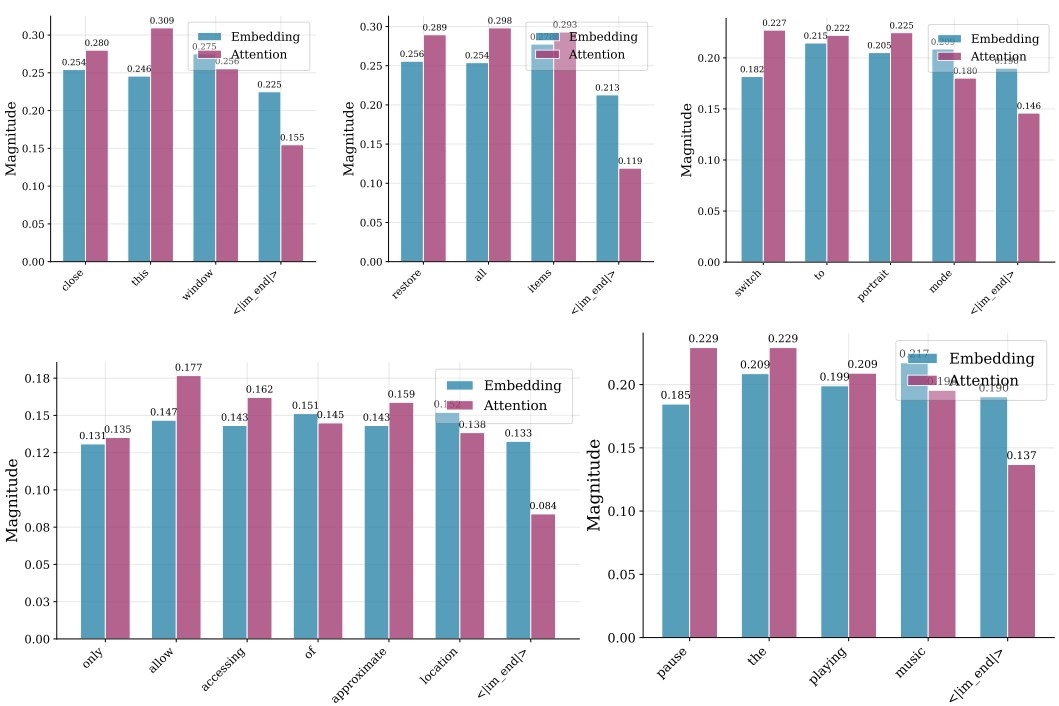

Figure 4: Magnitude of normalized visual correlation of query tokens computed from hidden states ("Embedding") and from multi-head self-attention ("Attention").

In Fig. 4, we compare normalized distributions of global visual-token correlation $\sum_{l=1}^{L} c_{q_i}^l$ computed form two different manners. "Embedding" denotes the $c_{q_i}^l$ computed from Eq. (7) based on hidden states. And "Attention" denotes visual-token correlation of $q_i$ with $c_{q_i}^l =$

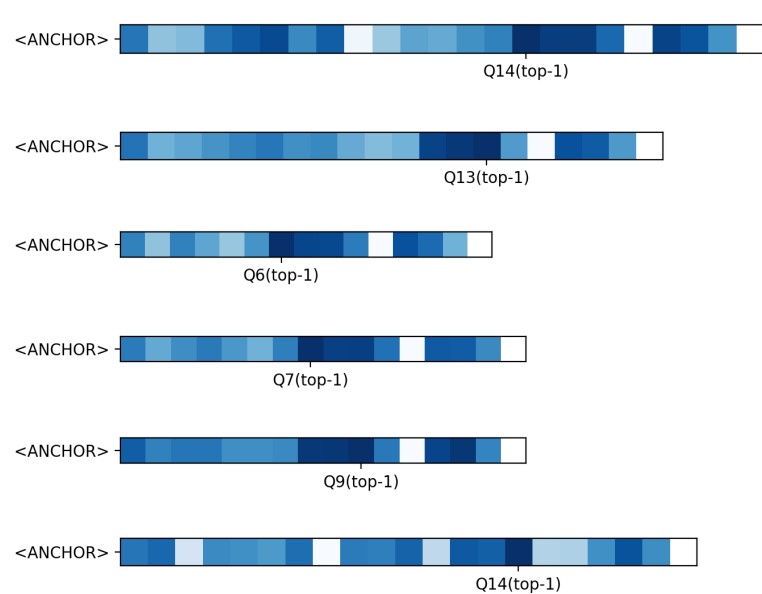

Figure 5: Cosine-similarity between multi-head weight vector computed from <ANCHOR> and each text token. The top-1 visual-query token consistently has the most similar multi-head weight vector with it computed from <ANCHOR>.

$\sum_{h \in [H]} \sum_{v_i \in \mathcal{V}} \mathbf{A}^{l,h}_{q_i, v_i}$ computed from multi-head attentions. From Fig. 4, we can observe different visual-correlation patterns: while the text token with largest magnitude in "Attention" manner sometimes falls in to semantic-irrelevant tokens, such as "this", "embedding" manner shows larger magnitudes on the `<|im_end|>` token.

The observations above verify the claim in paper that *the query-visual patten discovered in hidden states* $\mathbf{H}^l$ *is not necessary statistically prevailing among each head's self-attention matrix, as only a smaller subset of attention heads are "semantic heads" that key on semantic functionality and representation similarity*, which is also supported by Elhelo & Geva (2024); Olsson et al. (2022); Voita et al. (2019).

From ablation results in Section 4.2, the global pattern indicated from visual-sink query token computed $\mathcal{Q}_s$ from hidden states achieved better performance, support our selection that comply the visual-query pattern from hidden states for weighting attention grounding.

In Fig. 5, we compute the similarity values between the multi-head weight vector computed from <ANCHOR> token as Eq. (6) and computed from each query token $q_i$ as $w_{l,h} = \sum_{v_i \in \mathcal{V}} \mathbf{A}^{l,h}_{q_i, v_i}$ from the trained GUI-AIMA model. The top-1 visual-sink query token consistently has the most similar multi-head weight vector with it computed from <ANCHOR>, supporting our design that computing $w_{l,h}$ from visual-sink query token instead of <ANCHOR> token as it can work similarly with <ANCHOR> in computing multi-head weight vector but avoiding the bias from random initialized <ANCHOR> token.

# E    IMPLEMENTATIONS OF BASELINES

Most baselines' results are taken from the original papers, except GUI-G$^2$-3B which is not reported in the source.

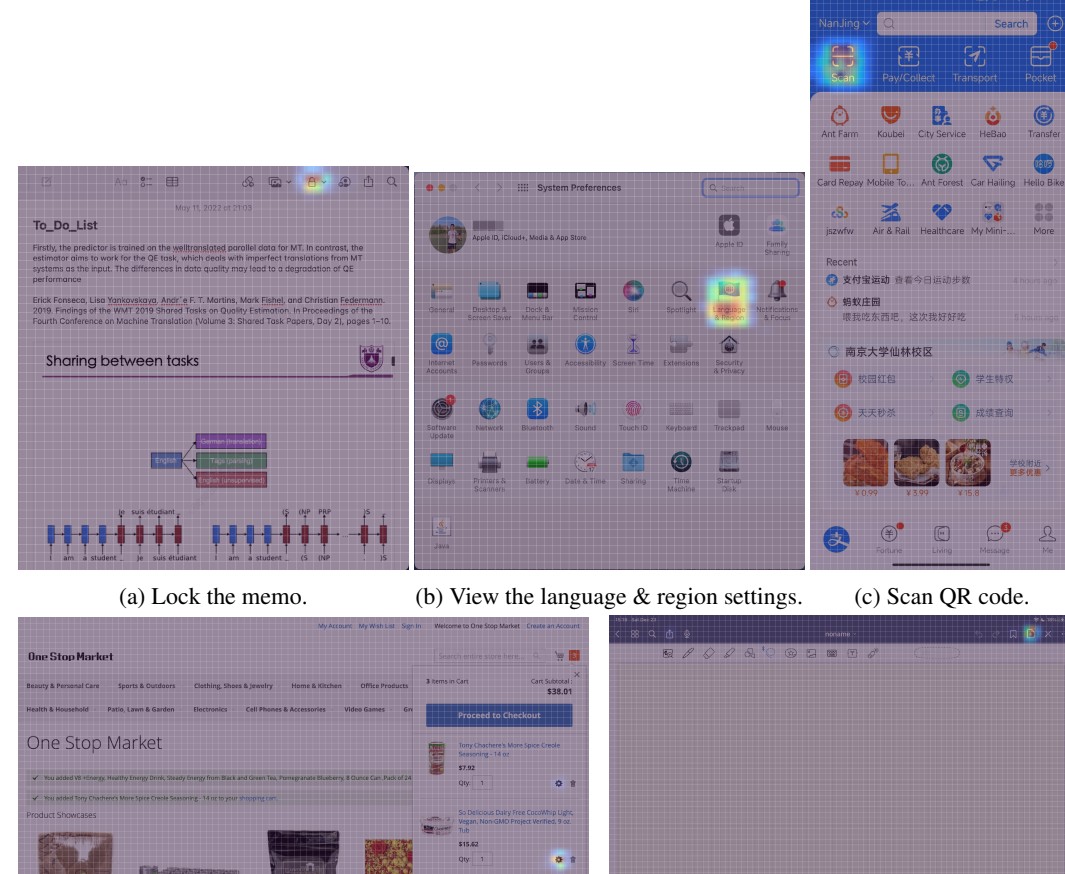

| (a) Lock the memo. | (b) View the language & region settings. | (c) Scan QR code. |

| (d) View settings of cocowhip light. | (e) Add a new page. |

Figure 6: Visualization examples of GUI-AIMA's grounding results on ScreenSpot-v2.

## F EXTRA DETAILS OF <ANCHOR> IMPLEMENTATIONS

In Section 3.3, we abbreviate the implementation of $[\mathcal{V}, \mathcal{Q}, \texttt{<ANCHOR\_START>}, \texttt{<ANCHOR>}, \texttt{<ANCHOR\_END>}]$ as $[\mathcal{V}, \mathcal{Q}, \texttt{<ANCHOR>}]$ for brevity. For the single-area prediction setting in GUI grounding, we explore to expand single <ANCHOR> to multiple <ANCHOR> tokens as $[\mathcal{V}, \mathcal{Q}, , \texttt{<ANCHOR\_0>}, , \texttt{<ANCHOR\_1>}, \ldots, \texttt{<ANCHOR\_N>}]$ (abbreviate start and end tokens). However, it turns out to bring no performance gains and merely adds redundancy for the single-region grounding tasks. We will explore the multi-region grounding tasks with disentangled <ANCHOR_n> token for each grounding region as future works.

## G GUI GROUNDING EXAMPLES OF GUI-AIMA

We provide the visualizations of GUI-AIMA's multi-head attention grounding results on ScreenSpot-v2 and ScreenSpot-Pro as followings in Fig. 6, Figs. 7 and 8.

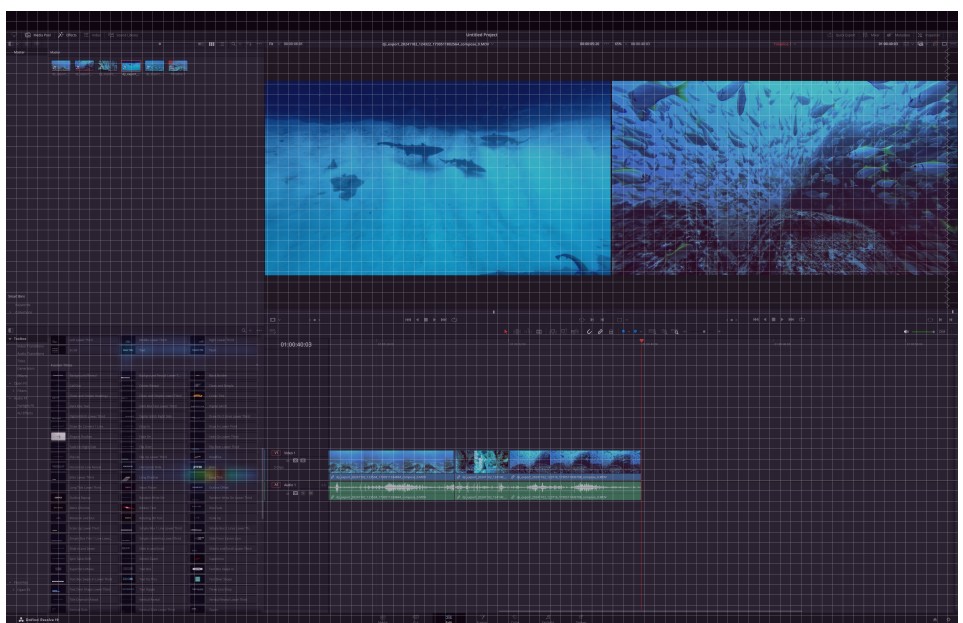

(a) Add long title.

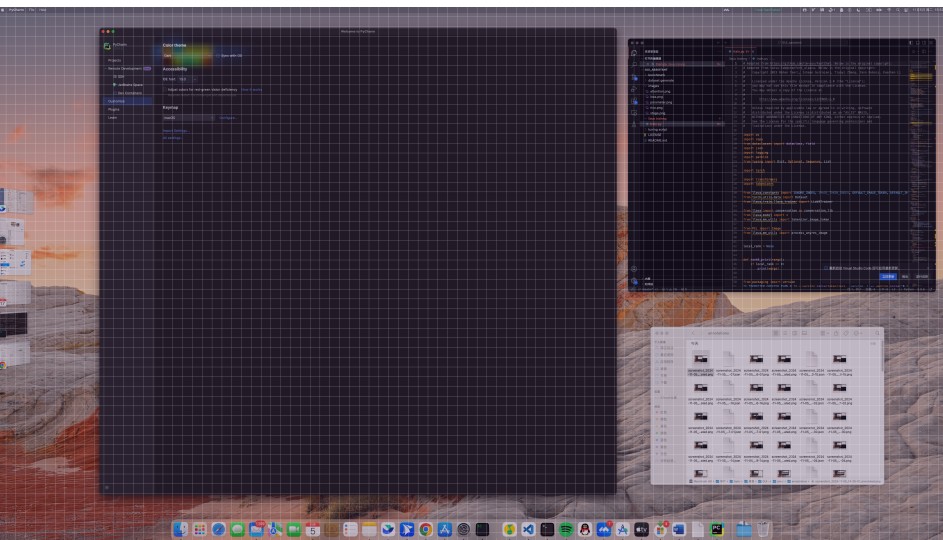

(b) Change color theme in pycharm.

Figure 7: Visualization examples of GUI-AIMA's grounding results on ScreenSpot-Pro.

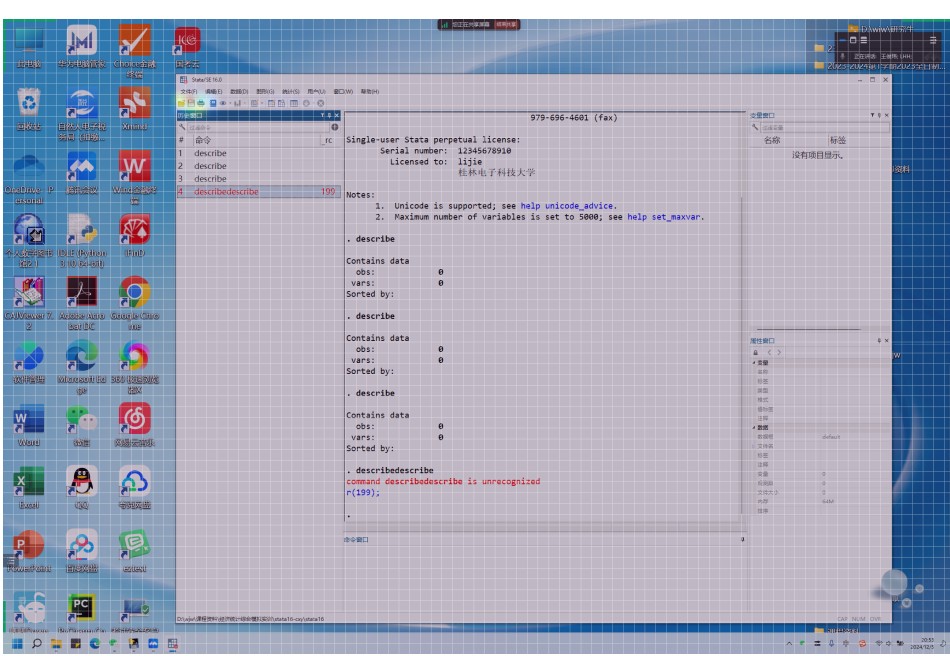

(a) Save a file.

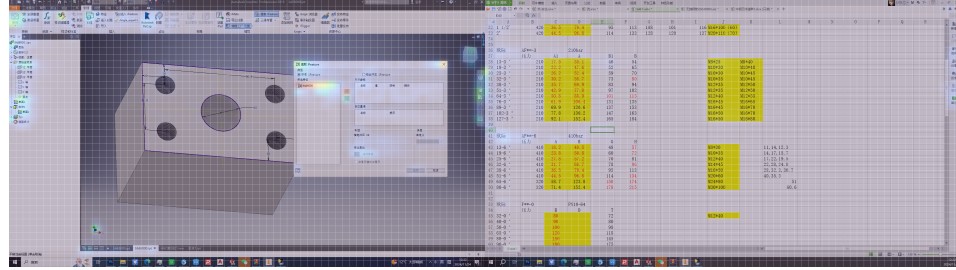

(b) Select the only feature.

Figure 8: Visualization examples of GUI-AIMA's grounding results on ScreenSpot-Pro.

