# OpenReview forum: "GUI‑AIMA: Aligning Intrinsic Multi-Modal Attention with a Context Anchor for GUI Grounding"
_ICLR.cc/2026/Conference — Submitted to ICLR 2026_

### Official Review · Reviewer_UHkz · 2025-11-01

**Soundness:** 3
**Presentation:** 3
**Contribution:** 2
**Rating:** 2
**Confidence:** 5

**Summary:**

This paper mainly focuses on the GUI Grounding problem and the authors explored utilizing intrinsic multimodal attention with a learnable context anchor token to propose a coordinate-free grounding method. To aggregate multi-head attention information, visual-query sink tokens are selected to generate weights for final attention-based grounding decision. Methods are verified on two popular GUI benchmarks to show the effectiveness.

**Strengths:**

Introducing the visual-sink query to construct attention-based grounding is interesting

**Weaknesses:**

1. L364-365 states that all model parameters were updated during training. Regarding the selection of Visual-sink Query Tokens in Equation (7), is this selection dynamic and layer-wise, changing with the input during the forward pass during training? Also, is a stop-gradient operation applied to the weight $w$ in Equation (4) or Equation (8)?
2. In Equation (4), the term $w_{l,h}$ is used to weight the anchor-visual token attention for the $l$-th layer and $h$-th head. However, L318-319 indicates that the specific value of $w_{l,h}$ is obtained by summing the attention scores between the selected visual-sink query token and the visual tokens. Given that the anchor-visual token relationship and the visual-sink query-visual token relationship are not necessarily the same, could the authors please provide a detailed justification for this direct substitution and usage? Is it possible to visualize the final selected visual-sink query token to illustrate which specific query token was chosen for grounding after model training?
3. In the experiments on ScreenSpot-v2 (Table 2), how were the performance metrics (avg: 90.4%) for GUI-Actor-3B obtained? According to Table 4 (bottom part) of the GUI-Actor paper, GUI-Actor-3B achieves 91.0% average accuracy, further improving to 92.4% with the addition of a verifier module.
4. Furthermore, the comparison against the GUI-Actor method appears to only consider a weakened version (i.e., the version without the verifier module). Given that the verifier module can be a common and general component for attention-based grounding methods, the authors should investigate if the proposed method could similarly benefit from the lightweight verifier module. Comparisons should ideally be made against the SoTA performance, rather than a potentially sub-optimal baseline.
5. The experimental results are exclusively based on the Qwen2.5-VL-3B-Instruct model. It is recommended exploring the performance of this method when scaled up to larger model sizes (like ~7B) to demonstrate its generalizability and effectiveness across different scales.
6. The paper currently only demonstrates the performance of GUI Grounding. As GUI grounding is merely one critical component of a comprehensive GUI Agent, the work lacks exploration into how the proposed method can be further extended to a functional GUI Agent System. Providing a demonstration of how this method contributes to a unified and efficient GUI Agent would be significantly more compelling and practical for the community.

**Questions:**

Please find the question in the Weaknesses section.

---

> ### Author Response · Authors · 2025-11-21
>
> We sincerely appreciate your valuable suggestions. Please see our clarifications below:
>
> **1.1. Selection of Visual-sink Query Tokens.** The selection is dynamic during training and inference for different input queries. As for whether layer-wise dynamic visual-sink query token, we conducted the ablations in **Table 4** and it shows that fixing the visual-sink query token through layers, as in Eq.9 achieves the best.
>
> **1.2. Stop-gradient on $w$.** There is no stop gradient on the multi-head weight for GUI-AIMA’s training. In **Table 7** of the Appendix, the ablations trained on the full training set, we show that the overall performance of GUI-AIMA-3B trained with stop-gradient on $w$ is overall worse, 4.05% worse on ScreenSpot-Pro, 1.95% worse on OSWorld-G, and 0.16% better on ScreenSpot-v2. These results indicate that amplifying the inherent query-visual functionality difference between attention heads is necessary without the stop-gradient on $w$.
>
> **2.** We agree with the reviewer that using the anchor-visual relationship reflected in each attention head, as in Equation 6 for multi-head weighting, is more intuitive and direct. However, during early training, the <ANCHOR> token embedding is premature. It has not yet learnt to represent the preceding query sequence, thus it will produce biased multi-head weights and mislead the training on the proper attention heads(lines 285-289). We show the ablation results of directly using anchor-visual-based weighing in **Table 4** (line “weighting with <ANCHOR> in Eq. (6)”), which performs similarly to uniform multi-head weighting that treats all attention heads equally. We also add the ablation of this variant trained on the full training set in **Table 7** of the Appendix, which is consistent with the results in **Table 4**.
> Our target is to emphasize attention heads with more focus on the cross-modal functionality, which are easier to learn a better anchor-visual attention vector for grounding. Since the <ANCHOR> token and other text tokens in the same attention head use the same QKV parameters, visual-sink query-visual relationships and anchor-visual relationship(with mature <ANCHOR> token) can both reflect the query-visual pattern of an attention head to form multi-head weights. And visual-sink query-visual relationships are accessible even in the early training, thus we form Eq.8 for weighting. We visualize the similarity between the multi-head weight vector from visual-sink query-visual relationships and the weight vector from anchor-visual relationships after the model is fully trained in **Figure 5 in Appendix D** with a detailed description (blue part),  where these two weight vectors share the most similarity.
> We also show the visualization of the differences between the query-visual patterns computed from hidden states and multi-head attentions, shown in **Figure 4** of the Appendix in the original submission.
>
> **3 and 4. GUI-Actor with verifier model.**
> (1) We re-evaluated the GUI-Actor-3B’s inference on the ScreenSpot-v2 and got the 90.4% result. In the updated manuscripts, we update them with the results with and without the verifier reported in the original paper.
> (3) In the **Table 6** of Appendix B, we show that GUI-AIMA also benefits from the verifier proposed by GUI-Actor and achieves the best performance with the verifier on ScreenSpot-v1 (89.9%), and ScreenSpot-v2 (93.0%), with the overall largest improvements. GUI-AIMA is also a patch-wise grounding method that can provide several candidate regions for the verifier to select, while GUI-Actor uses embedding similarity to form the map; GUI-AIMA directly utilizes the self-attention map. The results in **Table 6** indicate GUI-AIMA-3B not only provides a precise top-1 click point, but also gives more reliable top-k candidates.
>
> It should be noted that the extra stage of using verifiers will introduce non-ignorable extra time-consuming (e.g., ~63% extra time for ScreenSpot v1), and extra training of the verifier model. The verifier-based two-stage manner targets to use an extra module to correct the model's wrong preference on multiple candidate regions. Here, we supplement another two-stage manner for GUI-AIMA that focuses on fixing the offset errors of the original predicted single region without extra training (and module) in **Section 3.5**: crop and zoom-in the top-1 candidate region and pass it to the model to localize again. This crop then zoom-in extra stage is faster (partial image input) and training-free, especially useful on high-resolution screenshots, such as the samples in ScreenSpot-Pro and OSWorld-G. The extra time is much less: ~18% on ScreenSpot-Pro and ~32% on OSWord-G because the model only needs to do grounding on the crop instead of the entire screenshot in the second stage. The improvement is substantial, 49.8%->57.0% on ScreenSpot-Pro and 58.3%->63.8% on OSWorld-G.

---

> > ### Author Response · Authors · 2025-11-21
> >
> > **5. Different model size.** We trained a GUI-AIMA-7B with the same training set(85k screenshots with 254k elements) the results is as following:
> > | Model      | ScreenSpot-v2 Text | ScreenSpot-v2 Icon | ScreenSpot-v2 Avg. | ScreenSpot-Pro Text | ScreenSpot-Pro Icon | ScreenSpot-Pro Avg. | OSWorld-G Avg. |
> > |-----------|---------------------|--------------------|--------------------|----------------------|---------------------|---------------------|----------------|
> > | GUI-AIMA-3B | 97.21              | 84.12              | 91.51              | 62.03                | 29.97               | 49.78               | 58.33          |
> > | GUI-AIMA-7B | 96.52              | 86.46              | 92.14              | 64.38                | 30.79               | 51.55               | 61.70         |
> >
> > Due to the time limitation, this 7B variant is roughly trained without adapting any training configuration. We focus on the 3B model for efficient use cases, e.g., mobile GUI grounding.
> >
> > **6. Beyond grounding.** We focus on GUI grounding to demonstrate that the coordinate-free GUI grounding method can be much more data-efficient, and multi-head attention-based grounding is more fine-grained to achieve better grounding performance than the embedding-based method.
> > However, a unified GUI agent is a more natural and consistent practice to realize full functionalities: both planning and grounding. In **Appendix C: Future Work: Extending GUI-AIMA to Multi-Action and Multi-Element Grounding**, we add a detailed description of a unified manner of GUI-AIMA that plans the next action as an embedding-based classification via an extra classification head on the added <CLS> token’s final hidden states. After deciding the action type via the <CLS> token, it automatically adds a corresponding number of grounding <ANCHOR> tokens for grounding of the predicted action, which can be multi-element actions (e.g., drag-and-pull in a single iteration involves 2 regions). We leave it as future work for implementation due to the time limitation.

---

> ### Author Response · Authors · 2025-11-27
> **Gentle Reminder to Reviewer UHkz**
>
> Dear Reviewer UHkz,
>
> Thank you again for your detailed comments on our paper. In the rebuttal, we have carefully responded to your comments to address your concerns, and we would greatly value any feedback after reading our rebuttal. Please let us know if further information or clarification could help your assessment.
>
> Best, Authors

---

### Official Review · Reviewer_9cY1 · 2025-11-01

**Soundness:** 3
**Presentation:** 4
**Contribution:** 4
**Rating:** 6
**Confidence:** 4

**Summary:**

This paper proposes GUI-AIMA, an attention-only framework that leverages MLLMs’ intrinsic grounding capability via attention alignment. It does not rely on extra modules( e.g., GUI-Actor’s embedding-based heads) and also avoid cumbersome token aggregation and biased head selection(e.g., TAG) through their pipeline.

The main pipeline includes 3 major components:
1. It converts coordinate-based bounding boxes into soft labels by combining IoU (for overlap) and Gaussian distance. It can prioritize center patches (matching human click habits) and down-weights partial-overlap border patches, resolving the annotation gap between coordinates and patches.

2. Design a learnable <ANCHOR> token to the input sequence ([V, Q, <ANCHOR>]) to act as a surrogate for aggregating query-visual attentions. This eliminates the need for manual weighting of all query tokens (a flaw in TAG) and preserves MLLMs’ general capabilities by disentangling grounding from general understanding.

3. Identifies “visual-sink query tokens” to weight attention heads. Heads with robust text-vision affinity are prioritized, avoiding bias from premature <ANCHOR> tokens or irrelevant query tokens.

And the final result looks promising.

The ablation study is sufficient to support the claims of each module.

**Strengths:**

1. Novelty:
This paper proposed an efficient and effective pipeline for GUI grounding. It abandons traditional coordinate-based bounding box generation (which requires additional efforts like scaled corpora or OCR pretraining) and instead frames GUI grounding as patch-level attention prediction.

Also, the paper clearly motivates the work by highlighting limitations of existing methods. It proposes three novel components: overlap/center-aware patch labels (aligning with human click habits), visual-sink query tokens (for adaptive head weighting), and <ANCHOR>-based aggregation—each addressing a specific pain point in prior GUI grounding research.

The clarification of motivation and technical novelty looks good to me. Also, the theoretical details of each component are well demonstrated. These details ensure transparency in how the framework maps theoretical ideas to practical implementation.


2. Performance:
The result looks promising. Trained on only 85k screenshots (far less than SFT baselines requiring millions of samples), this proposed 3B model achieves state-of-the-art results among other 3B models. It even rivals larger 7B models and outperforms embedding-based (GUI-Actor-3B) and RL-based (GUI-G²-3B) baselines.

From this result, we could see that a well-designed small model for a specific domain might be able to compete with larger models. Also, the paper implicitly supports the potential of extending this framework to larger models. Its core designs are not limited to 3B backbones. Exploring broader visual grounding tasks, and scaling to larger models (with sufficient computing resources) could further enhance performance by leveraging more powerful pretrained capabilities while retaining GUI-AIMA’s efficient domain adaptation.

3. The paper is well-written and easy to catch.

**Weaknesses:**

1. The problem settings:

This work presents a strong contribution to addressing the GUI grounding problem through a well-designed coordinate-free pipeline. GUI grounding is indeed a critical component of GUI understanding, and the solutions proposed here hold practical value for advancing the development of GUI agents.

That said, the scope of GUI grounding itself is relatively narrow within the broader landscape of GUI agent capabilities. A potential concern is that enhancing grounding performance—even via the efficient attention-aligned framework proposed—might inadvertently compromise other core capabilities of the underlying MLLMs (e.g., reasoning), though this trade-off is not explicitly explored in the work.

2. Some details:
a.  The identification of visual-sink query tokens relies on cosine similarity with MLLMs’ intermediate hidden states, instead of attention matrices. It assumes hidden states consistently capture cross-modal affinity—a property that may not generalize to all MLLM backbones (e.g., models with different layer architectures or pretraining objectives).

b. I think the theoretical discussion is almost sufficient for me. Since it is for GUI agent, it would be better if the paper evaluates its performance in end-to-end agent workflows (e.g., combining grounding with action execution, handling dynamic GUI changes like pop-ups or resizing).

c. Some key hyperparameters (e.g., α=0.8 for Gaussian distance in patch labels, top-K=1 for visual-sink tokens) are set empirically without systematic analysis of their impact. The paper does not explore how varying these values (e.g., α=0.6/1.0, K=2/3) affects performance. Will those parameters change if we change to another dataset?

**Questions:**

As shown in the weakness part, the main questions are listed below:
1. For the problem setting, a key consideration arises: when the framework focuses exclusively on optimizing GUI grounding, will the MLLM’s capabilities for other purposes (e.g., reasoning or QA) be weakened? Additionally, when designing the entire pipeline, are there any trade-offs between the grounding task and other tasks that the MLLM is expected to perform?

2. Is this work easy to be generalized to other MLLM backbones considering the detailed design of calculating cosine similarity over hidden states.

3. Maybe some more practical discussion of this work and how it will contribute to the GUI agent.

4. As for the hyperparameters, will those parameters change if we change to another dataset?

---

> ### Author Response · Authors · 2025-11-21
>
> We sincerely thank the reviewer for the comments and suggestions. Please see our clarifications below.
>
> **1 and 3. Beyond GUI grounding.** As clearly mentioned in the title, this paper mainly focuses on designing effective and efficient attention-based grounding. And we show that the patch-wise attention-based grounding, with proper multi-head weighting, is much more data-efficient and more capable than previous coordinate-based AR methods and embedding-based methods (e.g., 49.8% versus 42.2% on ScreenSpot-Pro).
> However, although the decoupling of planning and grounding into two specialized models can satisfy the full functionalities of the GUI agent, unifying planning and grounding into a single model is more consistent and natural. A straightforward unified manner for GUI-AIMA is to first do action selection in natural language and then conduct grounding as the design. However, it is easy to induce multi-task conflicts between these two objectives: the action objective focuses on the language semantics of action texts, and the attention grounding objective focuses on the spatial precision.
> In **Appendix C: Future Work: Extending GUI-AIMA to Multi-Action and Multi-Element Grounding**, we add a detailed description of **an alternative unified manner** that treats action selection as an embedding-based classification via an extra classification head on the new <CLS> token’s final hidden states. It also enables GUI-AIMA to automatically realize the number of grounding areas required for the predicted action for multi-element grounding (e.g., drag-and-pull in a single iteration), and conduct grounding on the corresponding number of grounding <ANCHOR> tokens. We leave it as future work for implementation due to the time limitation.
>
> **2. Generalized to other MLLM backbones.** To identify the visual-sink query token, we use the final hidden states of each transformer layer, which are straightforward to access for different MLLM backbones. In addition to these hidden states, we need to extract the multi-head self-attention of the MLLM, which is unavailable when FlashAttention2 is enabled. To address this, we propose an efficient multi-head query–visual self-attention extraction method in “**Efficient Extraction of Self-Attention Map**” in the experiment section of the paper, which allows us to enable FlashAttention2 while still extracting the required attention. For different MLLM backbones, it only needs to adapt the query–visual attention extraction code (model_utils.py under /src/gui_aima in the supplementary codebase), which computes only the necessary partial attention maps rather than the full ones.
>
> **4. Hyperparameters.** When selecting the hyperparameters ($\alpha$ for labeling and $top-K$ for multi-head weighting), we rely on ScreenSpot-v2 (low-resolution dataset) and ScreenSpot-Pro (high-resolution dataset) as shown in **Table 4** in order to make the model generalize across screenshots with diverse resolutions. These two benchmarks also cover a wide range of GUI grounding scenarios. The $top-K$ selection is already included in **Table 4**. Here, we supplement the ablation results for $\alpha$ with the same training setting of **Table 4** to support our selection $\alpha=0.8$.
> | $\alpha$   | ScreenSpot-v2 | ScreenSpot-Pro |
> |-----------|---------------|----------------|
> | $\alpha=0.6$ |87.34%               |39.91%                |
> | $\alpha=0.7$ |89.15%               |41.62%                |
> | $\alpha=0.8$ |89.54%               |43.39%                |
> | $\alpha=0.9$ |89.47%               |42.88%                |
> | $\alpha=1.0$ |88.60%               |42.25%                |
>
> Our training data already includes diverse GUI grounding scenarios. And for the evaluation on different benchmarks, there is no need to change the hyperparameters and retrain the model. To support it, we add results of GUI-AIMA and baselines on OSWorld-G, an OS-centric benchmark whose overall resolution lies between ScreenSpot-v2 and ScreenSpot-Pro, in **Table 3 of Section 4**. GUI-AIMA-3B still achieves the best 3B performance and second-best result across all baselines on this benchmark.

---

> ### Author Response · Authors · 2025-11-27
> **Gentle Reminder to Reviewer 9cY1**
>
> Dear Reviewer 9cY1,
>
> Thank you again for your detailed comments on our paper. In the rebuttal, we have carefully responded to your comments to address your concerns, and we would greatly value any feedback after reading our rebuttal. Please let us know if further information or clarification could help your assessment.
>
> Best, Authors

---

### Official Review · Reviewer_FRCX · 2025-11-01

**Soundness:** 3
**Presentation:** 3
**Contribution:** 2
**Rating:** 4
**Confidence:** 4

**Summary:**

This paper proposes GUI-AIMA, a coordinate-free and attention-only framework for graphical user interface (GUI) grounding. Unlike previous methods that generate coordinates directly or add extra grounding modules, GUI-AIMA fine-tunes the intrinsic multimodal attention of multimodal large language models (MLLMs) to achieve efficient grounding without modifying the model architecture. Specifically, the method converts coordinate-based annotations into patch-wise soft labels that reflect overlap and center-click likelihood, and introduces a learnable <ANCHOR> token to aggregate attention across query tokens for simplified supervision. A visual-sink query token weighting mechanism further enhances grounding by emphasizing attention heads with strong cross-modal correlations while maintaining pretrained generalization capacity. Trained with only 85K screenshots, GUI-AIMA-3B achieves competitive performance among 3B-parameter models.

**Strengths:**

1. The paper proposes a clean, coordinate-free grounding framework that aligns multimodal attention without modifying the model architecture, offering a simple yet effective alternative to complex grounding modules.
2. Despite its simplicity, GUI-AIMA achieves competitive grounding performance with fewer training samples, and its attention-based formulation provides clearer interpretability of visual-textual alignment in GUI understanding.

**Weaknesses:**

*Method

The technical differences from GUI-Actor [1] are unclear. The paper positions GUI-AIMA as a coordinate-free alternative to prior grounding methods, but does not clearly explain how its attention mechanism fundamentally differs from that of GUI-Actor. Apart from removing the grounding verifier (referred to as the “extra adaptation stage” in this paper), the design seems conceptually similar since GUI-Actor also introduces an <ACTOR> token as a contextual anchor. It would be better to clarify what intrinsic attention alignment GUI-AIMA achieves that GUI-Actor’s attention schema cannot.

*Experiments

1. Missing key baselines. Although the paper discusses related methods such as GUI-Actor [1] and TAG [2], the experiments omit TAG from direct comparison. It is more convincing if considering it as an additional baseline.

2. Incomplete and possibly misleading performance comparison.

- The reported comparisons against GUI-Actor only use the variant without the grounding verifier. However, the GUI-Actor’s complete method consists of both the <ACTOR> token and verifier components. Even under this simplified baseline, Table 2 results show only comparable performance rather than a clear improvement (90.4 vs. 90.8). To make the performance claim convincing, it would help to either include results for the full GUI-Actor pipeline or evaluate on an additional dataset such as ScreenSpot-v1.

- Besides, since the paper argues the “extra adaptation stage” is inefficient, efficiency metrics (e.g., inference latency) are also necessary.

[1] Qianhui Wu, Kanzhi Cheng, Rui Yang, Chaoyun Zhang, Jianwei Yang, Huiqiang Jiang, Jian Mu, Baolin Peng, Bo Qiao, Reuben Tan, et al. Gui-actor: Coordinate-free visual grounding for gui agents. arXiv preprint arXiv:2506.03143, 2025.

[2] Hai-Ming Xu, Qi Chen, Lei Wang, and Lingqiao Liu. Attention-driven gui grounding: Leveraging pretrained multimodal large language models without fine-tuning, AAAI, 2025.

**Questions:**

Please refer to the weaknesses.

---

> ### Author Response · Authors · 2025-11-21
>
> We sincerely appreciate your insightful suggestions. Please see our clarifications below:
>
> **1. TAG baseline.** Since TAG is a training-free GUI grounding method, its performance (e.g., 54.8% on ScreenSpot-v1) is not directly comparable to SFT- or RL-based methods.In our paper, we refer to TAG’s grounding approach as **Vanilla Attention Grounding** (Section 3.1), and report its SFT results in the ablation of **Table 4**. Trained on the same 45k ablation set (line 361-364), GUI-AIMA reaches 42.13% on ScreenSpot-Pro, a +4.62% gain over TAG’s attention grounding (37.51%). We further compare convergence in **Figure 3**: both self-attention-based methods converge faster than the embedding-based GUI-Actor, but GUI-AIMA continues to improve without the mid-training plateau, indicating that uniformly weighting all attention heads suppresses head specialization and leads to a premature plateau.
> We add an extra comparison between **Vanilla Attention Grounding** (Eq.5) and GUI-AIMA trained on the full training set in **Table 7** of the Appendix: GUI-AIMA outperforms in all 3 benchmarks, consistent with the results in **Table 4**.
>
> **2. Comparisons with verifier.** (1) We retrained GUI-AIMA-3B with increased max_pixels of MLLM as described in general response 1 for fair comparison. GUI-AIMA-3B’s performance is improved: 49.8% on ScreenSpot-Pro, 91.5% on ScreenSpot-v2. GUI-AIMA-3B’s ScreenSpot-Pro performance is 7.6%, 5.2% better than GUI-Actor-3B and -7B with verifier. On ScreenSpot-v2, GUI-AIMA is 0.9% and 1.0% worse than GUI-Actor-3B and -7B with verifier, better than 3B model without verifier. We also add the result of ScreenSpot-v1 and OSWorld-G (the verifier model proposed in GUI-Actor does not work for OSWorld-G, decreasing the performance of both GUI-AIMA and GUI-Actor, not listed) into the Appendix. It should be noted that ScreenSpot-v2 is the re-annotated version of ScreenSpot-v1 with fewer annotation errors by OS-ATLAS [1], while others in v2 and v1 remain the same.
> (2) We also supplement the results of GUI-AIMA with the verifier on ScreenSpot-v1 and v2, with results shown in **Table 6** in Appendix B. GUI-AIMA-3B achieved 89.9% on ScreenSpot-v1 and 93.0% on ScreenSpot-v2, with overall more gains than GUI-Actor, indicating GUI-AIMA-3B not only provides preciser top-1 click point, but also gives more reliable top-k candidates.
>
> **3. Efficiency.** Under FlashAttention2, the training time of GUI-AIMA-3B and GUI-Actor-3B on the same training set (85k screenshots with 254k elements) is ~192 and ~260 A100 gpu hours. The average inference time of these two methods on ScreenSpot-Pro, on ScreenSpot-v2 and OSWorld-G is shown in the **Table 8** of the Appendix B. While the single inference time is similar (the extraction of partial self-attention map of all attention head consume the slight extra time as described in **Efficient Extraction of Multi-head Self-Attention** in Section 4) , GUI-AIMA-3B is more efficient on training due to the 1-stage training without extra warmup stage.
>
> [1] Wu, Zhiyong, Zhenyu Wu, Fangzhi Xu, Yian Wang, Qiushi Sun, Chengyou Jia, Kanzhi Cheng et al. Os-atlas: A foundation action model for generalist gui agents. arXiv preprint arXiv:2410.23218, 2024.

---

> ### Author Response · Authors · 2025-11-27
> **Gentle Reminder to Reviewer FRCX**
>
> Dear Reviewer FRCX,
>
> Thank you again for your detailed comments on our paper. In the rebuttal, we have carefully responded to your comments to address your concerns, and we would greatly value any feedback after reading our rebuttal. Please let us know if further information or clarification could help your assessment.
>
> Best,
> Authors

---

### Official Review · Reviewer_pAYT · 2025-11-02

**Soundness:** 3
**Presentation:** 2
**Contribution:** 2
**Rating:** 4
**Confidence:** 4

**Summary:**

The paper proposes GUI-AIMA, a coordinate-free, attention-only framework for GUI visual grounding, i.e., mapping a natural-language instruction to a clickable region on a graphical interface.
Unlike previous coordinate-based or embedding-based models (e.g., GUI-Actor), GUI-AIMA directly leverages the intrinsic multi-head self-attention (MHSA) within a multimodal large language model (MLLM). The method supervises attention weights with patch-wise soft labels converted from bounding boxes using an IoU-weighted Gaussian centered at the click point.
A learnable <ANCHOR> token is introduced to aggregate query-to-visual attentions, simplifying token-wise aggregation. Moreover, the paper proposes a visual-sink query token weighting scheme that adaptively emphasizes attention heads with strong cross-modal alignment.

While GUI-AIMA presents an elegant and efficient way to supervise intrinsic attentions for GUI grounding, its applicability is restricted to single-target tasks, and the conceptual novelty beyond simplifying previous attention aggregation is limited.
The idea is interesting and potentially impactful for broader multimodal grounding, but it currently lacks the depth and empirical diversity expected for acceptance at a major venue like ICLR.

**Strengths:**

- The idea of aligning the model’s own MHSA to grounding signals is appealing, and the proposed framework adds no extra modules and can be trained in a single stage. The <ANCHOR> token provides a neat mechanism for aggregating attention across query tokens.

- GUI-AIMA-3B outperforms all existing 3B-scale models and rivals much larger (7B–70B) models, showing high efficiency and good generalization with a modest dataset. The paper ablates key components (<ANCHOR> token, visual-sink weighting, and soft patch labeling), and results are consistent and interpretable.

**Weaknesses:**

- The proposed framework is only applicable to single-element grounding tasks and cannot handle dense and overlapping GUI grounding or multi-intent GUI agent tasks,  while prior GUI grounding methods like predicting coordinates as discrete language tokens in MLLM response can naturally perform multi-element grounding. There is no discussion about this limitation and potential ways to extend to complex GUI grounding scenarios.

- The visual-sink token weighting relies on heuristics (top-K cosine similarity), with limited justification. No quantitative analysis shows how these heads correlate with true cross-modal alignment. The paper would also strongly benefit from visualizing how <ANCHOR> attention maps change before/after training to support the claim of “aligning intrinsic attention.”

- The baselines mainly cover prior coordinate- or attention-based models. No comparison to new multi-stage grounding or reinforcement learning approaches that can act on multiple UI components.

**Questions:**

See weaknesses.

---

> ### Author Response · Authors · 2025-11-21
>
> We gratefully appreciate your efforts and suggestions. We would like to clarify and answer as follows.
>
> **1. deal with multi-element grounding.**
> Thank you for pointing this out. In this paper, we followed the previous single-element setting in the GUI grounding paper. However, in **Appendix C: Future Work: Extending GUI-AIMA to Multi-Action and Multi-Element Grounding**, we add a detailed method to enable GUI-AIMA to automatically realize the number of grounding areas of the predicted action and conduct grounding on the corresponding number of grounding <ANCHOR> tokens. We leave it as future work for implementation due to the time limitation.
>
> **2.1. Justifications of our multi-head weighting design.** We conduct an extra experiment to investigate whether our multi-head weighting can emphasize the helpful attention heads for GUI grounding.  We utilize the “weighting uniformly” checkpoint from **Table 4**. This checkpoint is trained with uniform multi-head weighting, i.e., all attention heads are treated equally without any learned preference. We verify the multi-head weighting of GUI-AIMA in Eq. (8) (with the visual-sink query token selected by Eq. (7)) by applying it at inference time on this “weighting uniformly” checkpoint. Specifically, we use the multi-head weights from Eq. (8) to select either the top 30% (172 heads) or the bottom 30% (172 heads) and drop all remaining heads’ predictions.
> Retaining the bottom 30% heads achieves 87.26% and 38.07% on ScreenSpot-v2 and ScreenSpot-Pro, which ruins the performance, while retaining the top 30% heads achieves 89.23% and 40.86%, respectively. These results indicate that the weighting mechanism of GUI-AIMA successfully identifies attention heads that capture the cross-modal correlations most relevant for GUI grounding.
>
> **2.2. Visualization.** As the <ANCHOR> token is randomly initialized, the attention map of the premature <ANCHOR> token cannot reflect meaningful patterns. Instead, we investigate the multi-head weighting vector from the <ANCHOR> token after training and find it is most similar to the weight vector that is used for model training computed from the visual-sink query token from **Eq.8**, as shown in **Figure 5 in Appendix D** with a detailed description. It supports our design that using a visual-sink query token to measure the cross-modal alignment of each head is a proper replacement of using <ANCHOR> as in **Eq.6**, that after training, the randomly initialized <ANCHOR> tokens can reflect a similar weighting pattern.
>
> **3. Baselines that can act on multiple UI components.**  We find two baselines: (1) SparkUI-Parser [1]: a method with improved parsing capability on multiple elements of the entire interface; (2) GMS [2], which is similar to patch-wise grounding methods that can generate multiple candidates, but during its multi-stage process, with an extra scanner. We add all their available results into the paper: ScreenSpot-v1 (**Table 5** in Appendix B), ScreenSpot-v2 (**Table 2**) results of SparkUI-Parser, ScreenSpot-Pro (**Table 1**) results of GMS (Qwen2.5-VL-7B results instead of proprietary model for fair comparison).
>
> [1] Jing, H., Chen, J., Rao, C., Dang, Z., Teng, J., Chu, T., Mo, J., Fang, S., Lin, H., Lv, R. and Ma, C., 2025. SparkUI-Parser: Enhancing GUI Perception with Robust Grounding and Parsing. arXiv preprint arXiv:2509.04908.
>
> [2] Li, Z., Song, G., Wang, Y., Xiong, Z., Yuan, J. and Cai, Y., 2025. Generalist Scanner Meets Specialist Locator: A Synergistic Coarse-to-Fine Framework for Robust GUI Grounding. arXiv preprint arXiv:2509.24133.

---

> > ### Author Response · Authors · 2025-11-27
> > **Gentle Reminder to Reviewer pAYT**
> >
> > Dear Reviewer pAYT,
> >
> > Thank you again for your detailed comments on our paper. In the rebuttal, we have carefully responded to your comments to address your concerns, and we would greatly value any feedback after reading our rebuttal. Please let us know if further information or clarification could help your assessment.
> >
> > Best,
> > Authors

---

### Author Response · Authors · 2025-11-21
**General response**

We sincerely appreciate all reviewers’ efforts and insightful reviews.

**1. Updated results.** GUI-AIMA-3B was trained with 30% less max_pixel (~4M) than GUI-Actor. We now retrain GUI-AIMA with the same max_pixel level of GUI-Actor while maintaining training data and other settings the same, and update the results in the manuscripts. The max_pixel setting decides the maximum visual patch tokens available for MLLMs, and directly affects the granularity of patch-wise GUI grounding.
In addition, we conduct evaluations on OSWorld-G and ScreenSpot-v1 in **Table 3 in section 4 and Table 5** of Appendix B. Specifically, GUI-AIMA-3B achieves 49.8% on ScreenSpot-Pro, 91.5% on ScreenSpot-v2, and 58.3% on OSWorld-G.

**2. Code and checkpoints.** We include the full code and checkpoints in the supplementary material for you to try.

**3. Ablations trained with full training set.** In **Table 7**, we add a new ablation study with all variant trained on the full training set (85k screenshots with 254k elements). It includes the results of uniform multi-head weighting with/without <ANCHOR> token, vanilla attention grounding weighted via Eq.5, GUI-AIMA-3B with stop-gradient on multi-head weights, GUI-Actor-3B with/without distance-aware labeling.

**4. 2-stage results with the verifier.** We add the GUI-Actor with the verifier to the main tables. Since GUI-AIMA-3B is also a patch-wise grounding method that produces several candidate regions, it also benefits from the verifier to select the most plausible one. We include the performance of GUI-AIMA-3B with the verifier in **Table 6** of Appendix B. GUI-AIMA-3B benefits more than GUI-Actor 3B and 7B and achieves the best performance on ScreenSpot-v1 and ScreenSpot-v2.

**5. 2-stage results with the crop and zoom-in.** For the patch-wise grounding method, besides the 2nd-stage verifier proposed by GUI-Actor, which focuses on selecting the precise regions from multiple candidates, we supplement another 2nd crop and zoom-in stage for GUI-AIMA introduced in **Section 3.5**, which is training-free and focuses on fixing the offset error inside the selected region. This second stage is helpful for high-resolution images, such as the screenshots in ScreenSpot-Pro and OSWorld-g. We add the results of GUI-AIMA with this second stage in **Table 1** and **Table 3**.

---

### Meta-Review · Area_Chair_Cwar · 2026-01-11

**Summary:**

A major recurring concern (Reviewers pAYT, UHkz) is that the proposed framework is restricted to "single-element grounding". In the current landscape of GUI agents, the ability to handle multi-element, dense, or overlapping grounding (and multi-intent tasks) is critical. The reviewers felt that coordinate-based methods handle this naturally, whereas the proposed attention-based method struggles without significant modification.

Reviewers FRCX and UHkz noted that the comparison with GUI-Actor was initially unfair because it compared the proposed method against a version of GUI-Actor without its "verifier" module. They argued that since the verifier is a general component, the proposed method should also be evaluated in that context or compared against the full GUI-Actor pipeline to prove superiority.

Reviewer FRCX pointed out that the conceptual novelty is limited, as it appears to be a variation of GUI-Actor (both use context anchor tokens) but without the verifier. Reviewer pAYT and UHkz questioned the heuristics used for the "visual-sink query token" weighting and the justification for substituting anchor-visual weights with query-visual weights.

Reviewers 9cY1 and UHkz noted that grounding is just one part of a GUI agent. The paper lacks an evaluation of how this module functions within a full agent system (end-to-end execution) or if optimizing for grounding compromises the MLLM's reasoning capabilities.

**Reviewer Concerns:**

There were no discussions taken place. The authors made efforts to address the missing baselines by adding SparkUI-Parser and GMS results. They also provided a comparison with the GUI-Actor + Verifier setup (Table 6) and provided efficiency metrics (Table 8), responding to Reviewer FRCX. The authors provided additional experiments (dropping top/bottom heads) and visualizations (Figure 5) to justify the multi-head weighting mechanism and the correlation between visual-sink tokens and learned anchor weights. This made progress addressing the technical justification concerns of pAYT. The authors provided ablation studies, addressing Reviewer 9cY1's concern about sensitivity.

While the authors added "Appendix C" outlining a theoretical approach to extend the work to multi-element grounding, they explicitly stated this is left for future work. This confirms the reviewers' concern that the current implementation is limited to single-target tasks, which is a significant limitation compared to coordinate-based approaches. The paper remains a study on a grounding module rather than a full agent system. The trade-offs between grounding optimization and general reasoning/planning capabilities remain unexplored. While the efficiency arguments are valid, the method essentially simplifies existing approaches (removing the verifier stage of GUI-Actor). While valuable, the performance gains over the full GUI-Actor pipeline (when both use verifiers) are marginal or mixed depending on the metric, making the "state-of-the-art" claim less decisive.

**Reviewer Scores:**

Reviewer pAYT (Current: 4) While the authors justified the weighting mechanism, the primary concern regarding the single-element limitation was acknowledged as "future work" rather than solved.

Reviewer FRCX (Current: 4): the authors addressed the unfair comparison and added efficiency metrics. However, the similarity to GUI-Actor and the fact that the method is positioned as a "simplified" version might prevent a higher score without a more distinct performance leap.

Reviewer 9cY1 (Current: 6) was already positive. The rebuttal addressed hyperparameter concerns, but the fundamental issue of "problem setting" (grounding vs. full agent reasoning) remains.

Reviewer UHkz (Current: 2) was very critical of the baselines. The authors provided the requested comparisons (with verifier) and preliminary 7B results. However, the reviewer's skepticism about the substitution of attention weights and the lack of a full agent system suggests they would remain negative.

---

### Decision · Program_Chairs · 2026-01-26

Reject